EMBO
Molecular Medicine

# Loss of AXIN1 drives acquired resistance to WNT pathway blockade in colorectal cancer cells carrying RSPO3 fusions

Gabriele Picco[1,2], Consalvo Petti[1,2], Alessia Centonze[2], Erica Torchiaro[1,3], Giovanni Crisafulli[1], Luca Novara[1], Andrea Acquaviva[4], Alberto Bardelli[1,2] & Enzo Medico[1,2,*] (iD)

## Abstract

In colorectal cancer (CRC), WNT pathway activation by genetic rearrangements of RSPO3 is emerging as a promising target. However, its low prevalence severely limits availability of preclinical models for in-depth characterization. Using a pipeline designed to suppress stroma-derived signal, we find that RSPO3 "outlier" expression in CRC samples highlights translocation and fusion transcript expression. Outlier search in 151 CRC cell lines identified VACO6 and SNU1411 cells as carriers of, respectively, a canonical PTPRK(e1)-RSPO3(e2) fusion and a novel PTPRK(e13)-RSPO3(e2) fusion. Both lines displayed marked *in vitro* and *in vivo* sensitivity to WNT blockade by the porcupine inhibitor LGK974, associated with transcriptional and morphological evidence of WNT pathway suppression. Long-term treatment of VACO6 cells with LGK974 led to the emergence of a resistant population carrying two frameshift deletions of the WNT pathway inhibitor AXIN1, with consequent protein loss. Suppression of AXIN1 in parental VACO6 cells by RNA interference conferred marked resistance to LGK974. These results provide the first mechanism of secondary resistance to WNT pathway inhibition.

**Keywords** colorectal cancer; gene fusion; R-spondin; targeted therapy; WNT pathway

**Subject Categories** Cancer; Digestive System; Systems Medicine

## Introduction

Most colorectal cancers display aberrant activation of the WNT pathway, leading to stabilization and nuclear translocation of β-catenin, a key transcriptional regulator controlling stem cell maintenance, proliferation, and differentiation (Krausova & Korinek, 2014). In a molecular survey carried out by The Cancer Genome Atlas, 93% of CRCs have been found to carry a genetic alteration in at least one of 16 WNT pathway genes, defining this pathway as a major driver of CRC (Cancer Genome Atlas, 2012). In particular, loss of function of the WNT pathway suppressor APC accounts for 70% of all CRCs. APC alterations typically occur at early steps of the colorectal adenoma–carcinoma sequence (Powell *et al*, 1992; Morin *et al*, 1997), a typical feature of "trunk" genetic events, present in all cancer cells and therefore therapeutically attractive (Swanton, 2012). Accordingly, established CRCs were found to critically depend on APC mutation-driven enhanced WNT signaling, even in the presence of additional cancer-driving mutations (Dow *et al*, 2015). Genomic rearrangements involving the RSPO2 and RSPO3 genes have been found to provide an alternative mechanism of aberrant WNT pathway activation in CRC (Seshagiri *et al*, 2012). These genes encode secreted proteins, R-spondins, that synergize with WNT ligands to promote β-catenin signaling (de Lau *et al*, 2011). RSPO2 and RSPO3 translocations are mutually exclusive with APC mutations and lead to aberrant expression of fusion transcripts in which the 5′ portion, upstream from the RSPO coding sequence, is typically contributed by the highly expressed EIF3E and PTPRK genes, respectively (Seshagiri *et al*, 2012).

In the case of the PTPRK-RSPO3 translocation, recent studies in patient-derived xenografts (PDXs) carrying the fusion transcript demonstrated that inhibition of WNT ligand secretion by porcupine blockade, or direct targeting of RSPO3 by antibodies, markedly inhibits tumor growth, promoting loss of cancer stem cell functions and differentiation (Chartier *et al*, 2016; Madan *et al*, 2016; Storm *et al*, 2016). These results highlight the clinical relevance of targeting RSPO3 rearrangements in CRC patients. As for all pathway-targeted therapies, availability of experimental models is crucial to characterize the dependency on pathway activators and to explore possible mechanisms of release from such dependency (Misale *et al*, 2014; Rosa *et al*, 2014). Indeed, mechanisms of acquired resistance to WNT pathway inhibition are currently unexplored. In this view, availability of established CRC cell lines spontaneously carrying

1 Candiolo Cancer Institute – FPO IRCCS, Candiolo, Torino, Italy
2 Department of Oncology, University of Torino, Candiolo, Torino, Italy
3 Istituto Nazionale Biostrutture e Biosistemi - Consorzio Interuniversitario, Roma, Italy
4 Department of Computer and Control Engineering, Politecnico di Torino, Turin, Italy
 *Corresponding author. Tel: +39 011 9933 234; Fax: +39 011 9933225; E-mail: enzo.medico@unito.it

RSPO3 fusions would provide a valuable resource. We recently reported a large collection of 151 CRC cell lines and associated global gene expression profiles, reliably representing the molecular heterogeneity of CRC (Medico et al, 2015). We have also shown that expression-based outlier analysis within this compendium leads to identification of therapeutically actionable oncogenic drivers (Medico et al, 2015). Considering that, in RSPO3 rearrangements, a strong promoter is placed upstream of the RSPO3 coding sequence, we hypothesized that searching for outlier RSPO3 expression might pinpoint cases carrying the respective translocation. This hypothesis was tested in RNA-seq-based expression data generated by the TCGA, checking the presence of the RSPO3 fusion in outlier cases. Outlier expression analysis was then extended to the 151 CRC cell lines, leading to the identification of two unique lines carrying a RSPO3 fusion. We employed these cell lines to investigate addiction to WNT pathway activation mediated by PTPRK-RSPO3 translocations, modeling primary sensitivity and acquired secondary resistance.

## Results and Discussion

### Identification of RSPO3 rearrangements in CRC samples by combined stroma/transcript expression analysis

PTPRK-RSPO3 rearrangements typically lead to aberrant expression of fusion transcripts by colorectal cancer cells that normally do not express RSPO3 (Sato et al, 2009; Seshagiri et al, 2012). Therefore, in principle, searching for CRC samples displaying high RSPO3 mRNA levels should pinpoint tumors bearing RSPO3 fusions. However, stromal cells have been identified as a source for RSPO3 expression in the intestine (Kabiri et al, 2014). To verify the stromal origin of RSPO3 transcripts in CRC, we exploited the fact that in CRC patient-derived xenografts (PDXs), human stroma is substituted by mouse stroma (Isella et al, 2015). Analysis of our previous species-specific analysis of RNA-seq data from CRC PDXs (Isella et al, 2015) revealed that the fraction of RSPO3 transcripts of mouse origin is 99.9%. As a consequence, it is expected that in a human CRC sample, RSPO3 mRNA levels are correlated with the richness of stroma that we found to be reliably approximated by a transcriptional cancer-associated fibroblast (CAF) score (Isella et al, 2015). We therefore analyzed the correlation between the CAF score and RSPO3 mRNA levels in a 450-sample human CRC RNA-seq expression dataset that we previously assembled from TCGA (Isella et al, 2015). As shown in Fig 1A, RSPO3 levels were highly correlated with the CAF score (Pearson correlation = 0.76), confirming that the expression of this gene in tumor samples is mostly due to stromal cells. However, among the samples expressing high RSPO3 levels (red rectangle in Fig 1A), some had low CAF score, suggesting a possible overexpression by epithelial cancer cells. Similar results were also obtained using additional stromal scores, namely the endothelial score and the leukocyte score (Appendix Fig S1). We therefore selected all cases with RSPO3 expression $Z$-score > 2 and searched for RSPO3 fusions in the corresponding TCGA RNA-seq data. A canonical PRPTK(ex1)-RSPO3(ex2) fusion transcript was detected in 6 out of the 14 selected samples (43%). To test whether RSPO3 fusions are enriched in samples with low stroma, for each of the 450 TCGA samples, we subtracted the CAF score

from the RSPO3 expression value and found that in a subset of samples, presenting low CAF score values and high RSPO3 expression, this delta value raises to $Z$-score levels usually observed for outlier genes (Fig 1B). The relationship between RSPO3 levels and CAF score was further validated in an independent 2140-sample CRC microarray dataset, highlighting the presence of samples with high RSPO3 expression not justified by abundant stroma (Appendix Fig S2). In principle, RSPO3 fusions are expressed only by cancer cells; therefore, outlier samples characterized by low CAF score should be enriched in fusion transcripts. To test this hypothesis, the 14 RSPO3 overexpressing samples described above were subdivided in two groups, respectively, having a positive and a negative delta (RSPO3-CAF score). Notably, the six RSPO3 fusions were exclusively present in the positive delta group, also corresponding to delta $Z$-score values higher than 2.5 (Fig 1B, Appendix Table S1). To verify whether additional RSPO3 fusions were present in samples not identified by the above approach, we extended the search for RSPO3 fusions to 12 additional TCGA RNA-seq samples expressing lower levels of RSPO3. In particular, for any expression level, we selected samples with the lowest CAF score, so that at least a fraction of the RSPO3 reads could theoretically derive from epithelial cells (Appendix Fig S3). None of these samples was found to carry a RSPO3 fusion, suggesting that our approach saturated the dataset and captured all the samples carrying RSPO3 rearrangements. Samples with lower levels were not explored because the limited number of expected RNA-seq reads covering RSPO3 would anyway not allow detection of reads covering a fusion transcript. Cross-check with an available pan-cancer database of fusion transcripts in TCGA samples (http://www.tumorfusions.org; Yoshihara et al, 2015) revealed that no additional RSPO3 fusions were detected in the 450-sample TCGA dataset used for our analysis.

These results show that the presence of RSPO3 rearrangements in CRC can be anticipated by high RSPO3 mRNA levels not justified by abundant stroma.

### Transcriptional outlier analysis identifies CRC cell lines carrying canonical and novel PTPRK-RSPO3 rearrangements

To search for established cell lines carrying RSPO3 rearrangements, we analyzed RSPO3 expression levels in our previously described compendium of 151 CRC cell lines (Medico et al, 2015). While, as expected, RSPO3 expression was low or absent in most cells, two lines, VACO6 and SNU1411, were found to markedly overexpress RSPO3 (Fig 1C). Notably, both cell lines are wild type for APC, but have a different molecular makeup: VACO6 are microsatellite instable and BRAF mutated (V600E); SNU1411 are microsatellite stable and KRAS mutated (G12C). These mutations render both cells insensitive to EGFR-blockade by cetuximab (Medico et al, 2015) and therefore attractive models for alternative targeted therapy approaches. The two lines have also different in vitro growth properties (Fig 1C, photo inserts): while SNU1411 form adherent colonies (Ku et al, 2010), VACO6 cells, established from a poorly differentiated cecal cancer, grow as aggregates in suspension (McBain et al, 1984). RNA-seq analysis of the VACO6 and SNU1411 transcriptomes revealed a canonical PRPTK(ex1)-RSPO3(ex2) fusion transcript in VACO6 (Fig 1D) and a novel PRPTK(ex13)-RSPO3(ex2) fusion transcript in SNU1411 cells (Fig 1E). RNA-seq results were further confirmed by PCR followed by Sanger sequencing

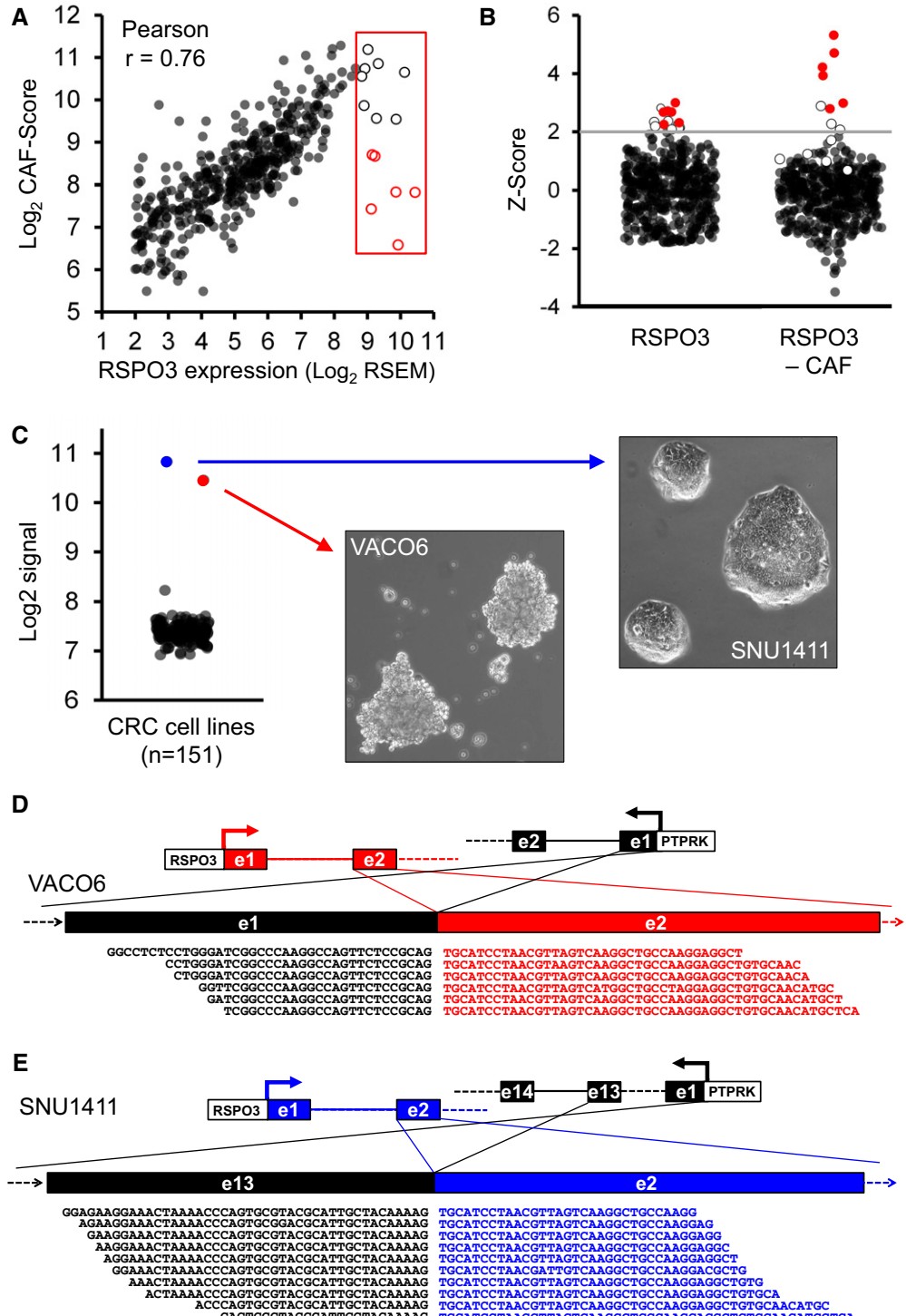

**Figure 1.   Identification of RSPO3 rearrangements in CRC samples and cell lines by outlier expression analysis.**

A   Scatter plot displaying the correlation between RSPO3 expression (*x*-axis) and CAF score (*y*-axis). The red box highlights samples with high RSPO3 expression and variable CAF score. Empty dots indicate samples selected for fusion analysis. Empty dots with red border indicate samples with high RSPO3 levels and medium-low CAF score.

B   Dot plots displaying the distribution of *Z*-score values for RSPO3 alone (left panel) and RSPO3 minus CAF score (right panel) in the 450-sample TCGA dataset. Red dots indicate RSPO3 fused samples. Empty dots indicate analyzed samples that do not carry fusions in the RSPO3 gene.

C   Scatter plot displaying RSPO3 expression levels (Log$_2$ signal, *y*-axis) in 151 CRC cell lines (left). Right images represents SNU1411 and VACO6 cell lines phase-contrast micrographs.

D   Canonical in-frame gene fusion between exon 1 of PTPRK and exon 2 of the RSPO3 gene revealed by RNA-seq analysis of VACO6 cells.

E   Novel in-frame gene fusion between exon 13 of PTPRK and exon 2 of the RSPO3 gene revealed by RNA-seq analysis of SNU1411 cells.

(Appendix Fig S4). The absence of reads covering exon 1 of RSPO3 confirmed that all RSPO3 transcripts detected in these cell lines originate from the fused transcript. These results highlight VACO6 and SNU1411 cell lines as unique cell line models to characterize addiction to WNT pathway activation by rearranged RSPO3 not only *in vivo*, but also *in vitro*, in the absence of supporting stroma.

### Cell lines carrying RSPO3 fusion transcripts are sensitive to porcupine inhibition *in vitro* and *in vivo*

The vast majority of CRC are affected by loss-of-function mutations in components of the destruction complex (e.g., APC) leading to accumulation of β-catenin and constitutive activation of Wnt target genes. RSPO3 instead promotes WNT pathway activation by binding the LGR4/5 protein and neutralizing RNF43-mediated degradation of LRP5/6 receptor, enhancing therefore the activity of WNT ligands (de Lau *et al*, 2014). Therefore, pharmacological blockade of WNT ligand secretion could be an effective strategy to target RSPO3-overexpressing cancer cells (Madan & Virshup, 2015). To this aim, we considered the small molecule LGK974 that specifically inhibits the porcupine (PORCN) acyltransferase, thus abrogating posttranslational processing and secretion of WNT ligands (Krausova & Korinek, 2014). LGK974 is currently being tested in humans, in phase 1 trials (Liu *et al*, 2013). VACO6 and SNU1411 cells were tested for *in vitro* dose–response to LGK974 and found to be both exquisitely sensitive, with $IC_{50}$ values below 50 nM (Fig 2A). As a control, HCT116 cells, that do not carry RSPO3 rearrangements, were insensitive to PORCN inhibition ($IC_{50} > 5$ μM). As shown in Fig 2B, both cell lines responded to LGK974 with marked apoptotic cell death and downregulation of the WNT pathway, evaluated by quantitative reverse transcription PCR (qRT–PCR) analysis of the WNT target gene AXIN2 (Drost *et al*, 2015; Jho *et al*, 2002; see Materials and Methods). To evaluate *in vivo* sensitivity of VACO6 and SNU1411 cells to WNT pathway inhibition, immunocompromised mice were xenotransplanted and treated with LGK974 or vehicle for 4 weeks. Xenotransplants of both cell lines responded to LGK974 with sustained growth inhibition (> 90%) and tumor stabilization (Fig 2C and D). Accordingly, tumors explanted at the end of the treatment displayed dramatic reduction in proliferating cells, and mucinous differentiation (Fig 2E and F), confirming that the *in vivo* response of both cell lines to WNT blockade phenocopies the described differentiation and growth arrest observed in CRC patient-derived xenografts (Storm *et al*, 2016). Altogether, these results show that both CRC cell lines carrying RSPO3 fusions are addicted to WNT signaling and sensitive to the porcupine inhibitor LGK974 *in vitro* and *in vivo*. Moreover, both models completely recapitulate morphofunctional traits of response previously described for RSPO3 blockade (Madan & Virshup, 2015; Storm *et al*, 2016). This is particularly interesting for SNU1411 cells with the noncanonical RSPO3 fusion, in which the long upstream coding sequence from PTPRK does not seem to lessen the pathological activation of WNT pathway driven by aberrant RSPO3 expression.

### AXIN1 frameshift deletions confer acquired resistance to WNT pathway inhibition in RSPO3-addicted cells

VACO6 cells, carrying the most common RSPO3 fusion, were subjected to long-term treatment with incremental doses of LGK974 (see Materials and Methods), which was found to generate

secondary resistance within 3 months. In the meantime, parental cells were maintained in culture without drug as a control. A viability assay demonstrated that selected cells were completely resistant to LGK974 (IC50 > 10 μM), while control parental cells maintained their sensitivity (Fig 3A). Copy number analysis based on exome sequencing of LGK974-resistant and parental VACO6 cells only highlighted minor changes of no clear functional meaning: trisomy of chromosome 8 and heterozygous deletion from 13q21.39 to 13q31.1. Moreover, a series of additional indels/mutations with high allelic frequency were detected (Appendix Table S2). In the context of a MSI cell line, a large set of mutations at low allelic frequency is expected as a consequence of genetic drift during expansion. However, in this case, the number of concurrent mutations at high allelic frequency points to the occurrence of a separated clone, pre-existing to the selection process. Interestingly, exome sequencing revealed that LGK974-resistant VACO6 (VACO6$_R$) cells diverged from the parental population by the presence of two single-base deletions in the AXIN1 coding sequence (Fig 3B, Appendix Fig S5), respectively, at position p.G265fs* (33 supporting reads) and p.V835fs* (90 supporting reads). Both deletions are reported in the COSMIC database (Forbes *et al*, 2015) as cancer-related somatic variants (COSM1609260 and COSM2920077) and the first one is predicted to be a truncating alteration (Cerami *et al*, 2012). Indeed, both mutations, detected as heterozygous in VACO6$_R$ cells, induce a frameshift in the coding sequence, which is compatible with a functionally homozygous AXIN1 loss of function. Accordingly, Western blot analysis showed a dramatic reduction in AXIN1 protein expression in VACO6$_R$ cells (Fig 3C).

To verify whether the two AXIN1 mutations co-existed in the same resistant clone, we isolated 14 independent clones from VACO6$_R$ cells by limiting dilution. All clones displayed complete loss of AXIN1 protein, as previously observed for VACO6$_R$ (Appendix Fig S6). Accordingly, we found that both frameshift mutations were present in all clones, each of them being heterozygous (Appendix Fig S6), which suggests that the vast majority of VACO6$_R$ cells emerged from one resistant subclone in which the two AXIN1 alleles were independently inactivated.

The AXIN1 protein acts as a scaffold for the destruction complex, in which GSK-3β phosphorylates β-catenin, leading to its ubiquitination and degradation. Therefore, loss of AXIN1 leads to stabilization of β-catenin and aberrant activation of the WNT pathway (MacDonald *et al*, 2009). Accordingly, VACO6$_R$ cells displayed enhanced mRNA expression of the WNT target AXIN2 (Fig 3D). AXIN1 and AXIN2 are considered to be functionally redundant (Chia & Costantini, 2005); however, AXIN2 is not always able to compensate for AXIN1 knockdown (Figeac & Zammit, 2015). Indeed, in the case of VACO6$_R$ cells, the observed AXIN2 upregulation does not effectively counteract the complete loss of AXIN1, which results in overall enhancement of WNT signaling.

To validate AXIN1 loss as a mechanism of secondary resistance to WNT pathway inhibition, we downregulated AXIN1 by RNA interference in parental VACO6 cells. Transient transfection of siRNAs against AXIN1 severely reduced AXIN1 protein expression in VACO6 parental cells (Fig 3E) and rendered them markedly resistant to LGK974 (Fig 3F). We verified by qRT–PCR that the RSPO3 fusion transcript was still expressed in VACO6$_R$ cells (Fig EV1A). Accordingly, in the mirror experiment, VACO6$_R$ cells were transduced with wild-type AXIN1 to levels comparable to parental cells

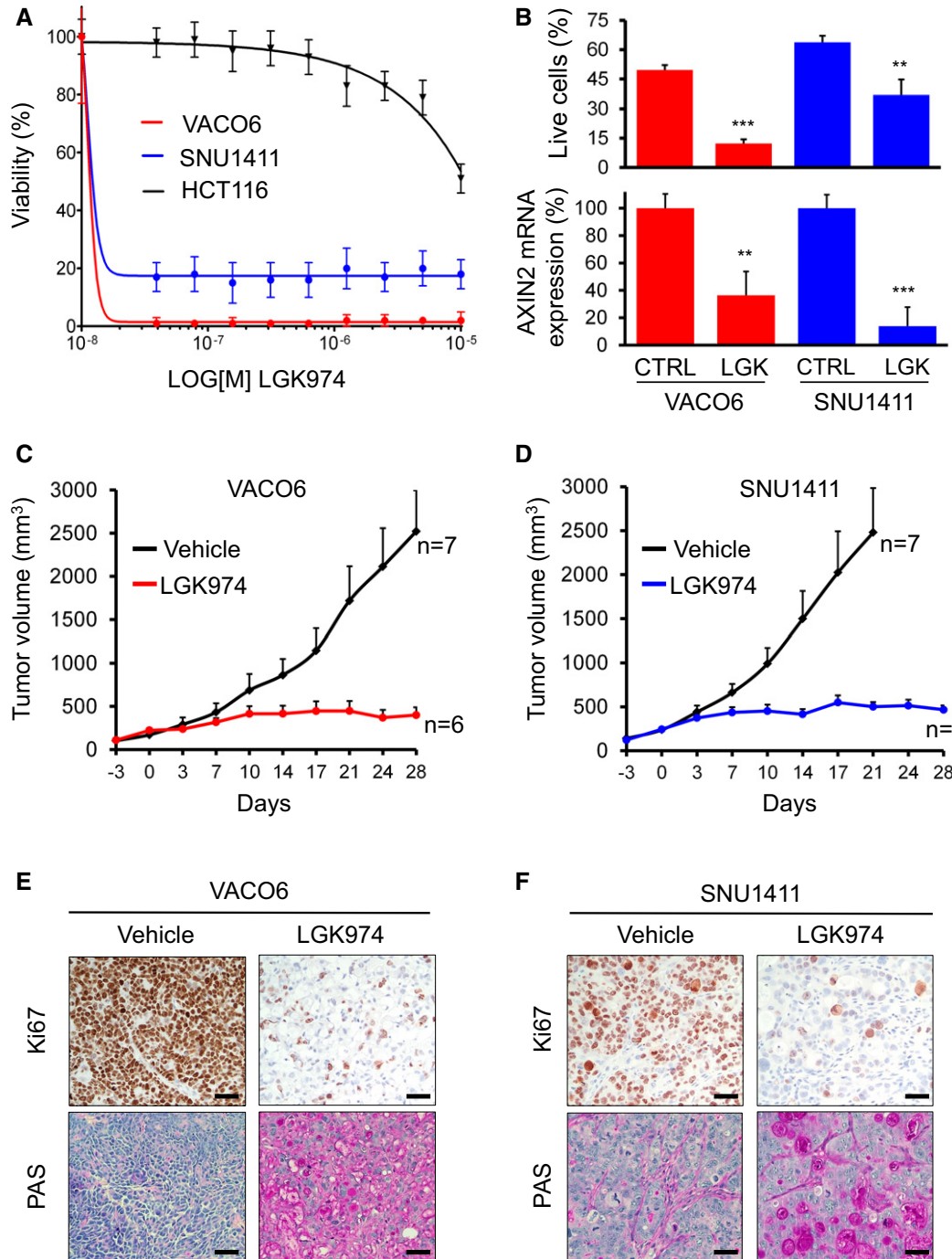

**Figure 2.  VACO6 and SNU1411 cells are sensitive to the porcupine inhibitor LGK974 *in vitro* and *in vivo*.**

A       Dose–response cell viability assay on VACO6, SNU1411, and HCT116 cells treated with LGK974. Data are expressed as average ± SD of six technical replicates from one representative experiment.

B       Effect of 1 μM LGK974 on apoptosis and WNT pathway activity in VACO6 and SNU1411 cells. Upper panel: percentage of viable cells (DAPI-negative & Annexin-negative; *y*-axis) after 96 h in control medium or 1 μM LGK974; means ± SDs from triplicate experiments. Asterisks represent significant differences measured by Student's *t*-test (two-sided), **$P < 0.05$; ***$P < 0.001$. VACO6 (LGK974 versus CTRL), $P = 3.88E{-}05$; SNU1411 (LGK974 versus CTRL), $P = 5.54E{-}03$. Lower panel: qRT–PCR evaluation of AXIN2 mRNA after treatment with vehicle or 1 μM LGK974 for 24 h, as indicated. Data are expressed as means ± RMSE of three technical replicates from one representative experiment measured by Student's *t*-test (two-sided), **$P < 0.05$ (0.006); ***$P < 0.001$. VACO6 (LGK974 versus CTRL), $P = 1.72E{-}03$; SNU1411 (LGK974 versus CTRL), $P = 6.12E{-}05$.

C, D    Tumor growth curves of PDXs from VACO6 and SNU1411 xenografts treated for 28 days with LGK974 (5 mg/kg; red line) or with vehicle (black line). Number of replicates: 6 or 7, as indicated in the panels. Error bars indicate standard error of the mean (SEM) values.

E, F    Micrographs of representative xenograft tumors derived from VACO6 (E) and SNU1411 (F) explanted at the end of treatment with vehicle or LGK974, stained with Ki67 or PAS, as indicated. Scale bar, 50 μm.

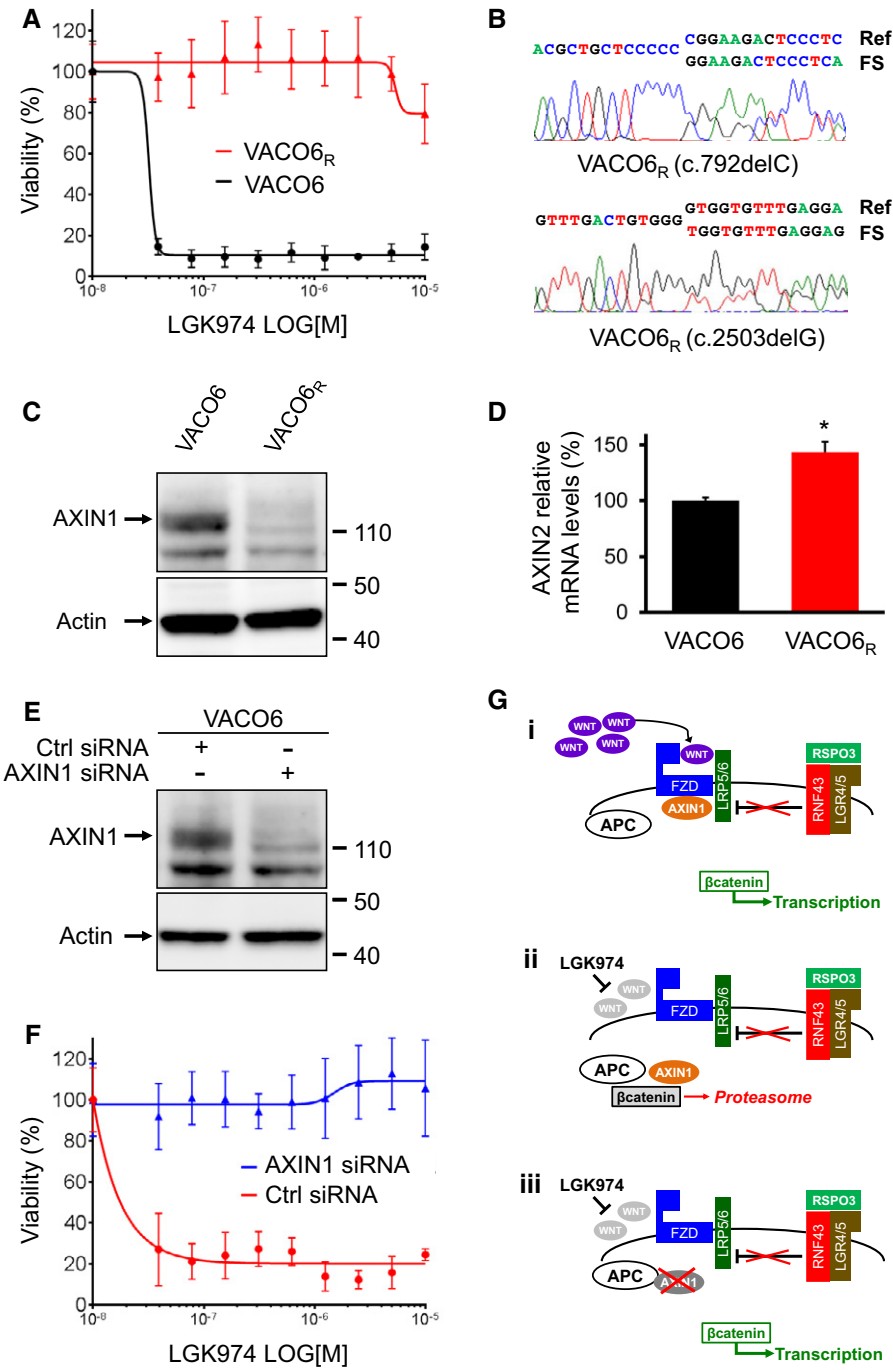

**Figure 3. AXIN1 loss confers acquired resistance to PORCN inhibition in VACO6 cell lines.**

A  Cell viability assay of VACO6 and VACO6$_R$ cells after 6 days of treatment with LGK974. Data are expressed as average ± SD of six technical replicates from one representative experiment.

B  Representative chromatograms of Sanger-sequenced PCR products confirming the two concurrent AXIN1 frameshift deletions (FS) in VACO6$_R$ cells.

C  Western blot showing loss of the AXIN1 protein in VACO6$_R$ cells.

D  Bar graph displaying relative mRNA levels of the WNT pathway target gene AXIN2, measured by qRT–PCR, in VACO6 and VACO6$_R$ cells. Two-sided Student's *t*-test, *$P < 0.05$, $P = 1.42E-02$. Data are expressed as means ± SD of three technical replicates from one representative experiment.

E  Western blot showing downregulation of the AXIN1 protein in VACO6 cells by RNAi.

F  ATP-based assay measuring viability, after 6 days of LGK974 treatment, of VACO6 cells transiently transfected with control siRNA or with AXIN1 siRNA, as indicated. Data are expressed as average ± SD of six technical replicates from one representative experiment.

G  Schematic representation of (i) how RSPO3 overexpression supports WNT-β-catenin signaling, (ii) how porcupine inhibition by LGK974 blocks β-catenin, and (iii) how AXIN1 loss restores β-catenin signaling in the presence of LGK974.

Source data are available online for this figure.

(VACO6$_R$AXIN1FL; Fig EV1B), which led to reduction in WNT signaling and reversion of resistance to LGK974 (Fig EV1C and D). These results further confirmed the causative role of AXIN1 loss in resistance to LGK974.

Genetic alterations leading to AXIN1 loss of function have been described in colon and other cancer types, albeit at low frequency (Satoh *et al*, 2000; Dahmen *et al*, 2001; Laurent-Puig *et al*, 2001; Wu *et al*, 2001; Cancer Genome Atlas, 2012; Forbes *et al*, 2015). In CRC, AXIN1 alterations typically occur between exon 1 and 5, where the binding domains for APC, GSK3, and β-catenin are located (Salahshor & Woodgett, 2005). In cell lines and transgenic mice, overexpression of AXIN1 leads to increased β-catenin degradation and attenuation of WNT signaling, supporting its tumor suppressor activity (Kishida *et al*, 1998; Hsu *et al*, 2001).

To verify whether AXIN1 loss confers resistance to additional WNT pathway inhibitors, we tested *in vitro* on VACO6 and VACO6$_R$ cells the alternative porcupine inhibitor WNT-C59 and the tankyrase inhibitor XAV939. While both WNT-C59 and XAV939 were effective on VACO6 parental cells, they had no effect on VACO6$_R$ cells (Appendix Fig S7A and B). To evaluate the pathway specificity of resistance in VACO6$_R$, we assessed their sensitivity to two chemotherapeutic agents commonly used to treat CRC patients, the antimetabolite 5-FU and the topoisomerase-I inhibitor SN38, and to Pevonedistat, a NEDD-8 inhibitor preclinically validated in CRC, to which parental VACO6 cells are markedly sensitive (Picco *et al*, 2017).

We found that parental and VACO6$_R$ cells display the same pattern of response to all three compounds (Appendix Fig S7C–E), indicating that AXIN1 loss confers resistance specifically to WNT pathway inhibition.

As illustrated in Fig 3G, here we show that, in CRC cells with RSPO3 fusions and wild-type APC, loss of AXIN1 by truncating mutations can sustain WNT pathway activity and resistance to more upstream pathway inhibitors, such as the PORCN inhibitor LGK974. These results confirm that loss of AXIN1 can sustain WNT signaling also in the presence of functional APC and in the absence of WNT ligands, providing the first genetic mechanisms of secondary resistance to WNT pathway inhibition. Indeed, the emergence of AXIN1 genetic inactivation after long-term treatment with porcupine inhibition of VACO6 cells strongly supports the idea that RSPO3-rearranged CRCs are critically dependent on WNT signaling. Moreover, as previously observed for the EGFR pathway (De Roock *et al*, 2010; Misale *et al*, 2014), both primary and secondary resistance to blockade of a central switch in a key signaling pathway can be achieved by genetic alterations affecting downstream nodes of the same pathway.

To further investigate whether AXIN1 inactivation is the only way for VACO6 cells to become resistant to LGK974, we transduced them with an activated β-catenin or induced transient APC down-regulation by RNA interference (Fig EV2A). In both cases, increased nuclear β-catenin levels (Fig EV2B) led to increased basal WNT pathway activation (Fig EV3A) and resistance to PORCN inhibition by LGK974 (Fig EV3B). These results confirmed that also alternative molecular alteration leading to WNT pathway activation, like APC loss and β-catenin activating mutations, can confer resistance to WNT pathway inhibition in CRC cells with RSPO3 rearrangements. Indeed, APC or β-catenin mutations are frequent in CRC and potentially expected as cause of primary and secondary resistance to

WNT pathway inhibitors. Conversely, AXIN1 alteration is less frequent and, in the clinical setting, would not be explored as first candidate to confer acquired resistance. In this context, AXIN1 loss could potentially drive also primary resistance to more upstream WNT pathway inhibitors.

## Materials and Methods

### Cell lines, drugs, and generation of resistant cells

HCT116 were purchased from American Type Culture Collection (ATCC), SNU1411 cells were purchased from Korean Cell Line Bank (KCLB), and VACO6 cells were provided by Prof. Markovitz (Cleveland, USA). All cell lines were maintained in their original culturing conditions according with supplier guidelines, as reported elsewhere (Medico *et al*, 2015). Cells were ordinarily supplemented with fetal bovine serum at different concentrations, 2 mM L-glutamine, antibiotics (100 U/ml penicillin and 100 mg/ml streptomycin), and grown in a 37°C and 5% $CO_2$ air incubator. Cells were routinely screened for the absence of mycoplasma contamination using the Venor GeM Classic kit (Minerva Biolabs). The identity of each cell line was checked by Cell ID System and by Gene Print 10 System (Promega). LGK974, WNT-C59, and SN-38 were purchased from Selleck Chemicals (Cat. No. S7143, S07037, and S4908, respectively), and the stock solution was prepared according to the manufacturer's guidelines. MLN4924 was obtained from Active Biochem (Cat. No.: A-1139), while XAV-939 and 5-FU were purchased from Sigma (Cat. No. X3004 and F6627).

VACO6 cells resistant to LGK974 were obtained by exposing parental cells to escalating doses of LGK974 until resistant population emerged (~3 months). In particular, cells were maintained in 50 nM LGK974 for 2 weeks, 100 nM for 2 weeks, 500 nM for 3 weeks, and subsequently maintained in 1 μM of LGK974. Activated β-catenin and 7TGP reporter plasmids were purchased from Addgene (https://www.addgene.org) Cat. No. 24313 and Cat. No. 24305, respectively. Plasmids were purified with Maxiprep kits (Invitrogen), and DNA concentration was measured by Nanodrop 1000. All lentiviral constructs were transfected into HEK293T cells plated in a 6-well plate at a density of $5 \times 10^4$ cells per well 1 day prior to transfection, using Lipofectamine 2000 (Invitrogen). Supernatants were harvested after 48 h, filtered through a 0.45-μm filter, and used to infect cells in the presence of 4 μg/ml of polybrene. After transduction, cells were selected in puromycin (2 μg/ml for 7 days).

### Gene expression datasets and stromal cell-specific signatures

To evaluate RSPO3 transcript expression in CRC samples, we exploited our previously assembled 450-sample TCGA mRNA dataset, available as Experiment Data package from Bioconductor: http://www.bioconductor.org/packages/release/data/experiment/html/TCGAcrcmRNA.html, as described elsewhere (Isella *et al*, 2015). To generate plots displayed in Appendix Fig S2, gene expression profiles from multiple datasets of CRC surgical specimens have been assembled in a unique large dataset and analyzed (see Data availability).

To accurately trace the stromal content from gene expression profiles of bulk CRC tumor samples, we exploited three stromal

signatures (for CAFs, leukocytes, and endothelial cells) that we previously developed from sorted CRC cell populations (Isella *et al*, 2015). Averaging the expression of genes in each signature yielded three stromal scores (CAF score, Leuco score, and Endo score, respectively), reporting the abundance of the three stromal cell populations in the sample.

### RNA extraction, RNA-seq library preparation, and analysis

Total RNA was extracted from SNU1411 and VACO6 using miRNeasy mini kit (Qiagen), according to the manufacturer's protocol. The quantification and quality analysis of RNA were performed on a 2100 Bioanalyzer (Agilent), using RNA 6000 nano Kit (Agilent). RNA-seq libraries were generated using Illumina TruSeq Stranded TotalRNA LT with Ribo-Zero Gold kit and validated using Bioanalyzer DNA 1000/High Sensitivity kit. Validated libraries were normalized to 10 nM and pooled in equal volume. 75-nucleotide-long single-end reads were performed on the NextSeq500 system following vendor's instruction.

### Detection of RSPO3 fusion transcripts

TCGA level 1 unaligned Illumina RNA-seq FastQ files were obtained from the Cancer Genomics Hub (https://browser.cghub.ucsc.edu). Bioinformatic analyses to define the presence of fusion transcripts in TCGA RNA-seq data were accomplished using Defuse (McPherson *et al*, 2011) and ChimeraScan (Iyer *et al*, 2011) for paired-end samples, while Mapsplice (http://www.netlab.uky.edu/p/bioinfo/MapSplice2) was used for single-end samples. GRCh37/hg19 genome was used as reference for alignments in all tools. Parameter settings are listed in Appendix Table S1B. Fusions in VACO6 and SNU1411 were identified using of a combination of BWA (Li & Durbin, 2010) and BLAT aligners (Kent, 2002). The reads potentially containing translocations were extracted from the alignment files and re-aligned using BLAT and then postprocessed to detect gene rearrangements. Fusion calling was performed with the following criteria: Each partner must have at least 25 mapped nucleotides on the respective end of the read; the two partners must map to two different genes.

### Exome sequencing and analysis

Libraries for exome sequencing were prepared with the Nextera® Rapid Capture Exome Kit (Illumina, Inc.), according to the manufacturer's protocol. Preparation of libraries was performed using 100 ng of genomic DNA from VACO6 and VACO6$_R$ cells, fragmented using transposons, adding simultaneously adapter sequences. Purified DNA was isolated after the fragmentation step and used as a template for subsequent PCR to introduce unique sample barcodes. Fragments' size distribution of the DNA was assessed using the 2100 Bioanalyzer with a High Sensitivity DNA assay kit (Agilent Technologies). The same amount of DNA libraries was pooled and subjected to hybridization capture. Libraries were then sequenced using the Illumina NS500 sequencer (Illumina, Inc.). FastQ files generated by Illumina NextSeq500 were preprocessed to remove all bases in the read with a Phred quality score less than 20. Sequences were mapped to the human reference (assembly hg19) using the BWA-mem algorithm bwa;

PCR duplicates were removed using the RMDUP command of SAMtools package (Li *et al*, 2009). Mutational discovery analyses were performed by a custom NGS pipeline, according to previously published methods (Siravegna *et al*, 2015), in order to call genetic variations in VACO6$_R$ respect to VACO6 cells, when supported by at least 1.5% allelic frequency and 5% significance level obtained with a Fisher test. To identify insertions and deletions (indels), we further analyzed the alignment files by comparing VACO6 and VACO6$_R$ cells using Pindel software (Ye *et al*, 2009). Mutations and indels were annotated by a custom script printing out gene information, number of normal and mutated reads, the allelic frequencies, and the variation effect. Each of these entries was associated with the corresponding number of occurrences in the COSMIC database (Forbes *et al*, 2015).

### Cell viability assay

CRC cell lines were seeded at different densities ($2–4 \times 10^3$ cells per well) in 50 μl complete growth medium in 96-well plastic culture plates at day 0. The following day, serial dilutions of LGK974 were added to the cells in additional 50 μl serum-free medium. Plates were incubated at 37°C in 5% $CO_2$ for 1 week, after which the cell viability was assessed by measuring ATP content through Cell Titer-Glo Luminescent Cell Viability assay (Promega). Luminescence was measured by Perkin Elmer Victor X4. For RNA interference experiments, cell viability was measured after 6 days of treatment.

### Apoptosis

To evaluate the fraction of live cells, SNU1411 and VACO6 were treated with 1 μM LGK974 for 96 h. Apoptosis was measured as staining with APC-conjugated Annexin V (Bender MedSystems, Burlingame, CA, USA) and DAPI (D9542, Sigma), in accordance with the manufacturer's instructions. The samples were analyzed on CyAN-Adp flow cytometer (Dako, Carpinteria, CA USA). Data acquisition was performed using Flow Jo Software.

### RNA interference

The siRNA-targeting reagents were purchased from Dharmacon, as a SMARTpool of four distinct siRNA species targeting different sequences of the AXIN1 transcript (L-009625-00-0005) and the APC transcript (L-003869-00-0005). Cell lines were grown and transfected with SMARTpool siRNAs using RNAiMAX (Invitrogen) transfection reagent, following manufacturer's instructions. Each experiment included the following controls: mock control (transfection lipid only), siControl1 (Dharmacon), AllStars (Qiagen) as negative control, and polo-like kinase 1 (Dharmacon), which served as positive control (Brough *et al*, 2011).

### qRT–PCR

Total RNA was extracted using the miRNeasy Mini Kit (Qiagen), according to the manufacturer's protocol. The quantification and quality analysis of RNA were performed by Thermo Scientific Nanodrop 1000 and 2100 Bioanalyzer (Agilent). DNA was transcribed using iScript RT Super Mix (Bio-Rad) following the manufacturer's instructions. qRT–PCR was performed in triplicate on ABI PRISM 7900HT

thermal cycler (Life Technologies) with SYBR green dye. The mRNA expression levels of the AXIN2 and the APC genes were normalized respect to PGK expression. Differential expression was statistically assessed by Student's *t*-test, although the limited group size ($n = 3$) does not allow reliable estimate of normality, which is commonly accepted for qPCR. The sequences of the primers (Sigma-Aldrich) used for gene expression analyses are as follows: AXIN2_FW, 5′-CGGGCATCTCCGGATTC-3′, AXIN2_RV, 5′-TCTCCAGGAAAGTTC GGAACA-3′, APC_FW 5′-GAAGGTCAAGGAGTGGGAGA-3′, APC_RV 5′-CTTCGAGGTGCAGAGTGTGT-3′, and PGK_FW, 5′-AGCTGCTGGG TCTGTCATCCT-3′, PGK_RV, 5′-TGGCTCGGCTTTAACCTTGT-3′.

### Western blot

Prior to biochemical analysis, $1.5 \times 10^6$ cells were grown in their specific media supplemented with 10% FBS. Total cellular proteins were extracted by solubilizing the cells in EB buffer (50 mM HEPES pH 7.4, 150 mM NaCl, 1% Triton X-100, 10% glycerol, 5 mM EDTA, 2 mM EGTA; all reagents were from Sigma-Aldrich, except for Triton X-100 from Fluka) in the presence of 1 mM sodium ortho-vanadate, 100 mM sodium fluoride, and a mixture of protease inhibitors. Extracts were clarified by centrifugation and normalized with the BCA Protein Assay Reagent kit (Thermo). Primary antibodies used were against AXIN1 antibody (Cell Signaling, C76H11—Cod.BK2087S) and anti-actin (Santa Cruz). Signals were revealed after incubation with anti-mouse or anti-rabbit secondary antibodies coupled to peroxidase (Amersham) by using enhanced chemilumi-nescence (ECL, Amersham).

### Animal models

Female NOD-SCID mice (Charles River Laboratories) were used in all *in vivo* studies. All animal procedures were approved by the Ethical Committee of the Institute and by the Italian Ministry of Health. The methods were carried out in accordance with the approved guide-lines. Nonobese diabetic/severe combined immunodeficient (NOD/SCID) male mice were purchased from Charles River Laboratories (Calco, Italy), maintained in hyperventilated cages, and manipulated under pathogen-free conditions. In particular, mice were housed in individually sterilized cages, every cage contained a maximum of 7 mice and optimal amounts of sterilized food, water, and bedding. VACO6 and SNU1411 xenografts were established by subcutaneous inoculation of $2 \times 10^6$ cells into the right posterior flank of 5- to 6-week-old mice. Tumor size was evaluated without blinding by caliper measurements, and the approximate volume of the mass was calculated using the formula $(d/2)2 \times D/2$, where d is the minor tumor axis and D is the major tumor axis. When tumors reached an average size of approximately 250 mm³, animals with the most homogeneous size were selected and randomized by tumor size. Vehicle or LGK974 (Cat. No. S7143; Selleck Chemicals), resuspended in 0.5% MC/0.5% Tween-80, were subcutaneously administered to mice 5 mg/kg daily. At least 6 mice for each experimental group were used to allow reliable estimation of within-group variability.

### Immunohistochemical staining

Formalin-fixed, paraffin-embedded tissues explanted from cell xeno-grafts were partially sectioned (10-μm thick) using a microtome.

### The paper explained

**Problem**

Colorectal cancer (CRC) is currently treated mainly by chemotherapy and, when possible, anti-EGFR targeted therapy. Recently, gene fusions involving the R-spondin family members RSPO2 and RSPO3 have been identified in CRC. These alterations promote WNT pathway activation and can be targeted by WNT pathway inhibitors. In this context, the understanding of the mechanisms of sensitivity and resis-tance to this new therapeutic strategy is limited by the lack of avail-ability of preclinical models featuring these specific alterations.

**Results**

By expression outlier analysis, we found two CRC cell lines, VACO6 and SNU1411, overexpressing RSPO3 and carrying, respectively, a canonical and a novel rearrangement of RSPO3. Both lines were extre-mely sensitive, *in vitro* and *in vivo*, to WNT blockade by PORCN inhi-bition. Long-term PORCN inhibition of VACO6 led to secondary resistance, driven by loss-of-function alterations of the AXIN1 gene, encoding a suppressor of WNT signaling.

**Impact**

The newly identified RSPO3-rearranged stable cell lines provide a useful preclinical model for characterizing sensitivity and resistance to WNT pathway inhibitors in CRC. As an initial exploitation of this model, we identified and validated AXIN1 genetic inactivation as the first described mechanism of secondary resistance to WNT pathway blockade.

4-μm paraffin tissue sections were dried in a 37°C oven overnight. Slides were deparaffinized in xylene and rehydrated through graded alcohol to water. Endogenous peroxidase was blocked in 3% hydrogen peroxide for 30 min. Microwave antigen retrieval was carried out using a microwave oven (750 W for 10 min) in 10 mmol/l citrate buffer, pH 6.0. Slides were incubated with mono-clonal mouse anti-human Ki67 (1:100; Dako) overnight at 4°C inside a moist chamber. After washings in TBS, anti-mouse secondary antibody (Dako Envision+System horseradish peroxidase-labeled polymer, Dako) was added. Incubations were carried out for 1 h at room temperature. Immunoreactivities were revealed by incubation in DAB chromogen (DakoCytomation Liquid DAB Substrate Chromogen System, Dako) for 10 min. Slides were counterstained in Mayer's hematoxylin, dehydrated in graded alcohol, and cleared in xylene, and the coverslip was applied by using DPX. A negative control slide was processed with secondary antibody, omitting primary antibody incubation. Immunohistochemically stained slides for Ki67 were scanned with a 20× objective, and representative images were been acquired. Periodic acid-Schiff (PAS) staining was purchased by Bio-Optica (Cat. No. 04-130802), and the staining was performed following the manufacturer's instructions.

### Immunofluorescence

Cells, grown on glass coverslip, were fixed in 4% paraformaldehyde for 20 min at room temperature and permeabilized with 0.1% Triton X-100 in PBS for 2 min on ice. Then, cells were treated at room temperature with 1% BSA in PBS for 30 min and incubated for 2 h at room temperature with the primary anti-β-catenin antibody (Purified Mouse Anti-β-Catenin, Cat. No. 610154, BD Transduction Laboratories™) diluted in PBS containing 1% donkey

serum. After washing, cells were fluorescently labeled, according to the primary antibody used, with anti-mouse-647 (A-21236, Thermo-Fisher) diluted 1:400 in PBS containing 1% donkey serum for 1 h. Nuclei were stained with DAPI. Coverslips were then mounted using the fluorescence mounting medium (Dako, Glostrup, DK) and analyzed using a confocal laser scanning microscope (TCS SPE II; Leica, Wetzlar, D) equipped with 63×/1.40 oil immersion objective.

### Flow cytometry

GFP expression analysis of *in vitro* cultured cells was performed by flow cytometry: Cells were trypsinized, diluted in a 1% paraformaldehyde-2% FBS solution, stained with DAPI (D9542, Sigma), and analyzed with FACS flow cytometer (CyAn™, DAKO).

### Data availability

The following datasets, available in the Gene Expression Omnibus (GEO) database, were used in this study: GSE59857 (Medico *et al*, 2015), GSE14333 (Jorissen *et al*, 2009), GSE35896 (Schlicker *et al*, 2012), GSE37892 (Laibe *et al*, 2012), GSE20916 (Skrzypczak *et al*, 2010), GSE17536 (Smith *et al*, 2010), GSE13294 (Jorissen *et al*, 2008), GSE39582 (Marisa *et al*, 2013), and GSE2109 (http://www.intgen.org/research-services/biobanking-experience/expo/). KFSYSCC was from https://www.synapse.org/#!Synapse:syn4974668.

Expanded View for this article is available online.

## Acknowledgements

We thank Carlotta Cancelliere, Roberta Porporato, Barbara Martinoglio, Daniela Cantarella, Michela Buscarino, Alice Bartolini, and Stefania Giove for technical assistance; Claudio Isella, Sara Erika Bellomo, Federica Invrea, Barbara Lupo, and Simona Lamba for help and advice; and Simona Destefanis for secretarial assistance. AIRC investigator grants (IG 16819 to E. Medico and IG 16788 to A. Bardelli), AIRC 9970-2010 Special Program Molecular Clinical Oncology 5x1000 to E. Medico and A. Bardelli; Fondazione Piemontese per la Ricerca sul Cancro 5x1000 Ministero della Salute 2010 and 2011 to E. Medico and A. Bardelli; European Community's Seventh Framework Programme under grant agreement no. 602901 MErCuRIC (A. Bardelli); IMI contract n. 115749 CANCER-ID (A. Bardelli); H2020 n. 635342-2 MoTriColor (A. Bardelli); QNRF National Priority Research Program n. 4-967-3-262 (E. Medico).

## Author contributions

GP contributed study design and data analysis and performed *in vitro* and *in vivo* experiments and manuscript writing. CP, AC, and ET performed *in vivo* and *in vitro* experiments. AA, GC, and LN performed bioinformatics analyses. AB contributed study design and manuscript writing. EM contributed study design, data analysis and bioinformatics, manuscript writing, and project oversight.

## Conflict of interest

The authors declare that they have no conflict of interest.

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
