## [Review Process File · EMBO Molecular Medicine]

Loss of AXIN1 drives acquired resistance to WNT pathway blockade in colorectal cancers cells carrying RSPO3 fusions

Gabriele Picco, Consalvo Petti, Alessia Centonze, Erica Torchiario, Giovanni Crisafulli, Luca Novara, Andrea Acquaviva, Alberto Bardelli, Enzo Medico

Corresponding author: Enzo Medico, University of Torino

Review timeline:

Submission date:	04 July 2016
Editorial Decision:	28 July 2016
Revision received:	23 November 2016
Editorial Decision:	06 December 2016
Revision received:	18 December 2016
Accepted:	20 December 2016

Transaction Report:

Editor: Roberto Buccione

1st Editorial Decision

28 July 2016

Thank you for the submission of your manuscript to EMBO Molecular Medicine. We have now heard back from the Reviewers whom we asked to evaluate your manuscript.

As you will see, the three reviewers all recognize the interest and importance of your work, but also raise significant, partially overlapping concerns. Although I will not dwell into much detail, I would like to highlight the main points

Reviewer 1 has one main concern, i.e. s/he wonders whether it is always axin or other Wnt pathway gene mutated in the resistant clones or if maybe other pathways substitute for can substitute for Wnt signaling. The reviewer suggests a comparative analysis between the outcomes of various selections to establish this point. You will also note that Reviewer 3 has the same fundamental concern, although s/he suggests different experimental approaches to address it. Reviewer 3 also lists a number of other items that require action, and which are important to consolidate the clinical relevance of your findings. Reviewer 2 is less reserved, but does ask for important clarifications on the Axin 1 mutations.

After further discussion with my colleagues and reviewer cross-commenting, we have decided to give you the opportunity revise your manuscript. As specifically for the shared concern by Reviewers 1 and 3, we suggest that 1) addressing the clonality of the 2 presented mutations might identify 2 unique clones which may help abrogate the need for lengthy repeat porcupine exposures and re-sequencing and 2) the functional experiments at points 4 and 5 proposed by Reviewer 3 would strengthen your claims and would be achievable in a shorter time frame.

In conclusion, while publication of the paper cannot be considered at this stage, we would be pleased to consider a revised submission, with the understanding that the Reviewers' concerns must be addressed with additional experimental data where appropriate and that acceptance of the manuscript will entail a second round of review.

It is important that you consider that it is EMBO Molecular Medicine policy to allow a single round of revision only and that, therefore, acceptance or rejection of the manuscript will depend on the completeness of your responses included in the next, final version of the manuscript.

As you know, EMBO Molecular Medicine has a "scooping protection" policy, whereby similar findings that are published by others during review or revision are not a criterion for rejection. However, I do ask you to get in touch with us after three months if you have not completed your revision, to update us on the status. Please also contact us as soon as possible if similar work is published elsewhere.

Finally, please note that EMBO Molecular Medicine now requires a complete author checklist (<http://embomolmed.embopress.org/authorguide#editorial3>) to be submitted with all revised manuscripts. Provision of the author checklist is mandatory at revision stage; The checklist is designed to enhance and standardize reporting of key information in research papers and to support reanalysis and repetition of experiments by the community. The list covers key information for figure panels and captions and focuses on statistics, the reporting of reagents, animal models and human subject-derived data, as well as guidance to optimise data accessibility.

We now mandate that all corresponding authors list an ORCID digital identifier. You may do so though our web platform upon submission and the procedure takes <90 seconds to complete. We also encourage co-authors to supply an ORCID identifier, which will be linked to their name for unambiguous name identification.

I look forward to seeing a revised form of your manuscript as soon as possible.

***** Reviewer's comments *****

Referee #1 (Remarks):

R-spondins stimulate Wnt signaling at the receptor level. Fusions of R-Spondins with other genes were previously shown to occur in colorectal cancer and are associated with increased R-spondin expression: The authors go the converse way by using high R-spondin expression as a marker to identify tumor samples and cell lines with R-spondin gene fusions. Inhibition of Wnt secretion blocked cell growth, and prolonged inhibition led to the emergence of resistant clones which exhibit de novo mutation of the downstream negative regulator Axin.

The approach taken is appealing and the experimentally data are sound and well described. However, with respect to novelty the impact of the story is rather limited. Mutations of R-spondin and sensitivity of R-spondin mutated tumor cells to inhibition of R-spondin or Wnt secretion have been described before. That escapers from inhibition show mutation in downstream regulators such as axin1 might have been expected. It would broaden the impact of the study if the outcome of several selections were compared in order to determine whether it's always axin (or another gene in the Wnt pathway) that is mutated in resistant clones, suggesting strong addiction of the cells to Wnt signaling, or whether other pathways are affected that substitute for Wnt signaling. Admittedly, such a broad scale analysis is costly but would allow more general statements as to resistance against Wnt signaling inhibition.

Minor points

Figure 3G suggests that R-spondin associates with LRP5/6 receptors, which is wrong. The consensus is that R-spondin acts by binding to Lgr5 and RNF43, thereby inhibiting the function of RNF43E3 ligase in removing Wnt receptors from the surface.

In the axin scheme in appendix Figure 4 the parts labelled "AXI.." were not clear.

Referee #2 (Comments on Novelty/Model System):

This paper nicely exhibits how novel bioinformatic analysis can be applied to publically available data and linked to relevant functional work to generate clinically translatable and important findings.

The only weakness in the study is with the evaluation of the acquired Axin1 mutations as detailed below

Referee #2 (Remarks):

This paper uses a logical but novel bioinformatic technique to strip out stromal derived signal from publically available RNA seq data to identify tumours with high epithelial expression of R-Spondin. This is then correlated with the acquisition of R-Spo fusion protein mutations in these tumours. This is a highly relevant finding as these mutations may not be detected with standard exome sequencing yet are associated with a characteristic molecular pathway. Importantly, the ligand dependent nature of Wnt activation in these tumours has been shown to be sensitive to Porcupine inhibition.

The paper then goes on to identify 2 cancer cell lines with previously undetected R-Spo fusion protein mutations and shows that they are sensitive to porcupine inhibition, but that escape from this inhibition can occur through the acquisition of other wnt disrupting mutations in Axin1

It is a very nice demonstration of how applied bioinformatics with functional follow up can maximally utilise publically available data to generate clinically translatable and relevant results. I like the paper very much

Specific comments and recommendations

Major

I only have 1 major comment - the identified "escape" Axin1 mutations are not completely characterised or considered

1. The c2503delG mutation looks to me as though it may actually be a c.2499delT mutation. This could be clarified with single strand cloning (using PGEMT or something similar). A c2499delT mutation is not listed in COSMIC, although this doesn't necessarily matter.

2. The identified mutations appear to be in polyC and poly G repeat tracts, was there any evidence of acquired microsatellite instability to explain this? Polymerase slippage could also be excluded with single strand cloning

3. Any copy number changes in the VACO6R cells?

4. The authors have not completed single cell/clone analysis to determine whether these 2 widely spaced frameshift mutations are occurring in the same escaper cell clone. If this is the case why would a cell need to acquire 2 frameshift mutations when the first will suffice to truncate the protein? This should be considered.

If the alternative scenario is true, and two different populations of the same escaper line have independently acquired a different Axin1 mutation this would be strong evidence for positive selection and an important functional role of this protein in Porcupine inhibition escape. Why this gene and not other frequently mutated Wnt disrupting mutations such as APC, Ctnnb1 should be considered and commented on.

Minor comments

How the CAF score was generated should be in the methods. It is too big a part of the story to require referral to a separate paper

Page 2, fig S1. Is the correlation of RSpO expression with the leucocyte and endothelial score because these scores are simply surrogate markers of the volume of tumour stroma. Or do the endothelium and leucocytes act as a source of R-Spo?

Page 2 Any idea what causes the high epithelial R-Spo score in the other 8/14 tumours without

detected R-Spo fusions?

Page 3 'in' needed - and a novel PRPTK(ex13)-RPO3 (ex 2) fusion transcript IN SNU1411

Page 3 - last paragraph

A sentence setting out the differences between the ligand dependent Wnt activation caused by R-SPO fusions as apposed to the constitutive activation caused by APC for example, would be useful for the non-expert reader

Page 4 para 1 -previously described for RSPO3 blockade. Not referenced

Page 4 para 1 - Last sentence isnt very clear. Do they mean that the long upstream coding sequence from PTPRK does not lessen the pathological activation of R-SPO3 expression?

Figure 2C and D Label the cell line on the figure

Referee #3 (Remarks):

Comments on manuscript number: EMM-2016-06773

Title: Loss of Axin1 drives acquired resistance to WNT pathway blockade in colorectal cancers cells

A small, but clinically nonetheless relevant subgroup of colorectal cancers carries RSPO2/3 fusion genes that lead to aberrant activation of Wnt/ β -catenin signaling in a manner dependent on the availability of Wnt ligands. Cancers driven by RSPO2/3 fusion gene expression therefore provide a unique opportunity for therapeutic intervention through the inhibition of Wnt ligand production or function. The design and realization of such therapeutic approaches could be fostered enormously by the development of suitable in vitro models. As a first step towards this goal Gabriele Picco and co-workers develop an elegant bioinformatic approach that should allow for high confidence identification of colorectal tumor samples which show high level tumor-specific expression of RSPO3 and which harbor RSPO3 fusion genes. This approach is based on the processing of transcriptome data and involves filtering out interfering gene expression signals from contaminating stromal cells in order to identify outlier samples with RSPO3 overexpression. The major achievement of this study is the successful application of the "outlier expression" approach to identify two colorectal cancer cell lines with PTPRK-RSPO3 fusions. These cell lines are further established as valuable preclinical model systems by demonstrating in vitro and in vivo their sensitivity against an inhibitor of Porcupine, an acyltransferase which is essential for Wnt ligand production. The potential utility of the newly identified cell lines is additionally demonstrated through the identification of mutations in the AXIN1 gene that might be the underlying cause for acquired resistance against the Porcupine inhibitor used. This manuscript is well written, is easy to follow, and provides sufficient background information to convey the significance and the importance of the study also to a non-expert readership. The technical quality is high and the vast majority of the authors' conclusions are strongly supported by the analyses and data presented. In my opinion, there are only two aspects that require further elaboration: These are, first, the reliability of the identification process of cancers with RSPO2/3 fusion genes, and second, the potential mechanisms whereby cancers with RSPO2/3 fusions might acquire resistance against Wnt pathway inhibitors, specifically the role of AXIN1 mutations is in this context.

Specific comments

1. By looking at Figure 1A and Appendix Table 1A it appears to me that one might have identified tumor samples with RPSO2/3 fusions without further bioinformatic analysis simply by picking the samples with high RSPO3 expression and a low CAF score. I suggest to individually label the 14 tumor samples within the red box in Figure 1A such that they can be tracked throughout the manuscript for example in Figure 1B, Appendix Figure 1A and Appendix Figure 1B. This would help to answer the question whether the same tumor samples are identified after stratification using the endothel and leukocyte gene expression signatures. Furthermore, are there false negatives, i. e. tumor samples with RSPO2/3-fusions that were not identified through the data processing pipeline developed by the authors?

2. The difference between the RSPO3 and the (RSPO3-CAF) Z-score for the GSE14333 data set is much less pronounced and clear-cut (Appendix Figure 2) when compared to the analysis of the TCGA data set shown in Figure 1B. So what is the significance of this analysis for the identification with high confidence of tumor samples with RSPO2/3 fusions? Which of the GSE14333 samples are actually positive for RSPO2/3 fusion genes? If this information is not available for GSE14333, how would the analysis for the tumors samples described by Seshagiri and co-authors (Nature 488, 660-664, 2012) look like?
3. The authors ponder whether the long upstream portion from the PRPTK gene in the PTPRK(ex13)-RSPO3(ex2) fusion interferes with RSPO3 function. But does this fusion protein actually exist or is it somehow processed to release the RSPO3 portion? This could be addressed by Western blotting to analyze the expression of the RSPO3 fusion protein and its apparent molecular weight.
4. From the manuscript it does not become clear whether the AXIN1 mutations are the only newly acquired genetic alterations in VACO6R cells. This needs to be clarified because if not, the significance of AXIN1 inactivation as resistance mechanism would be dubitable despite the phenocopy by the AXIN1 knockdown. Along the same line, can re-expression of AXIN1 restore sensitivity to LGK-974? Furthermore, it should be clearly shown that the RSPO3-fusion is still present and expressed in LGK-974-resistant cells.
5. After dose escalation for three months the authors isolate a LGK-974-resistant cell population and report the biallelic inactivation of AXIN1 in this population. Is there something special and perhaps unique about AXIN1 with respect to resistance against Porcupine inhibition? This could become important for the development of drugs that break secondary resistance. The authors need to provide information whether their resistant cell population is of monoclonal or polyclonal origin. Did AXIN1 inactivation occur repeatedly and independently in multiple cases? Is AXIN1 inactivation the only way to become insensitive towards LGK-974? The authors should knockdown APC and overexpress mutant β -catenin in VACO6 cells and compare this to AXIN1 knockdown with respect to LGK-974 sensitivity.
6. From a therapeutic point of view it would be interesting and highly relevant to know whether AXIN1 mutation/knockdown confers resistance specifically against LGK-974. How about resistance against other Porcupine and Wnt pathway inhibitors? Did VACO6R cells become resistant against other drugs, unrelated to the Wnt pathway, as well?
7. The authors show that AXIN2 is expressed in VACO6 cells and that AXIN2 is further upregulated in VACO6R cells. AXIN1 and AXIN2 are considered to be functionally redundant and fully interchangeable. In view of this, how do the authors explain that elevated AXIN2 does not compensate the loss of AXIN1? What are the expression ratios of AXIN1:AXIN2? Is AXIN2 expressed at the protein level?
8. The title should be more precise and avoid any kind of generalization that could be misleading. It should be clearly mentioned that the study applies to colorectal cancers with RSPO2/3 fusion genes and the interference with pathway activity at the level of Wnt ligand production/action.
9. The Chartier et al 2015 and Madan et al 2015 references are incomplete in the bibliography.

1st Revision - authors' response

23 November 2016

Point by point reply**Referee #1 (Remarks):**

R-spondins stimulate Wnt signaling at the receptor level. Fusions of R-Spondins with other genes were previously shown to occur in colorectal cancer and are associated with increased R-spondin expression: The authors go the converse way by using high R-spondin expression as a marker to

identify tumor samples and cell lines with R-spondin gene fusions. Inhibition of Wnt secretion blocked cell growth, and prolonged inhibition led to the emergence of resistant clones which exhibit de novo mutation of the downstream negative regulator Axin.

The approach taken is appealing and the experimentally data are sound and well described.

However, with respect to novelty the impact of the story is rather limited. Mutations of R-spondin and sensitivity of R-spondin mutated tumor cells to inhibition of R-spondin or Wnt secretion have been described before. That escapers from inhibition show mutation in downstream regulators such as axin1 might have been expected. It would broaden the impact of the study if the outcome of several selections were compared in order to determine whether it's always axin (or another gene in the Wnt pathway) that is mutated in resistant clones, suggesting strong addiction of the cells to Wnt signaling, or whether other pathways are affected that substitute for Wnt signaling. Admittedly, such a broad scale analysis is costly but would allow more general statements as to resistance against Wnt signaling inhibition.

→ We thank the Reviewer for his/her general comments and appreciation of the work. Regarding the potential impact of the work, also according to remarks and suggestions by Reviewer#3 and by the Editor, we verified that also APC loss or beta-catenin activation, when exogenously induced, may confer resistance to porcupine inhibition. Despite mutation in downstream regulators of WNT pathway might have been expected, this is the first work describing such escaping mechanisms in unique preclinical models suitable to model primary sensitivity and acquired resistance to an innovative therapeutic strategy currently explored in clinical trials. In particular, while APC or B-catenin mutations are frequent in CRC and potentially expected as cause of primary and secondary resistance to WNT pathway inhibitors, AXIN1 alteration is instead rare and, in the clinical setting, would not be explored as first candidate to confer acquired resistance. In summary, we think that this work strongly support the importance of AXIN1 as negative modulator of WNT signaling pathway in a potentially clinically relevant context.

Minor points

- Figure 3G suggests that R-spondin associates with LRP5/6 receptors, which is wrong. The consensus is that R-spondin acts by binding to Lgr5 and RNF43, thereby inhibiting the function of RNF43E3 ligase in removing Wnt receptors from the surface.

→ We have now modified Figure 3G accordingly.

- In the axin scheme in appendix Figure 4 the parts labelled "AXI.." were not clear.

→ We have now modified Appendix Fig S5. Legends for color-coded rectangles were included to specify the AXIN1 protein domains.

Referee #2 (Comments on Novelty/Model System):

This paper nicely exhibits how novel bioinformatic analysis can be applied to publically available data and linked to relevant functional work to generate clinically translatable and important findings. The only weakness in the study is with the evaluation of the acquired Axin1 mutations as detailed below

Referee #2 (Remarks):

This paper uses a logical but novel bioinformatic technique to strip out stromal derived signal from publically available RNA seq data to identify tumours with high epithelial expression of R-Spondin. This is then correlated with the acquisition of R-Spo fusion protein mutations in these tumours. This is a highly relevant finding as these mutations may not be detected with standard exome sequencing yet are associated with a characteristic molecular pathway. Importantly, the ligand dependent nature of Wnt activation in these tumours has been shown to be sensitive to Porcupine inhibition.

The paper then goes on to identify 2 cancer cell lines with previously undetected R-Spo fusion protein mutations and shows that they are sensitive to porcupine inhibition, but that escape from this inhibition can occur through the acquisition of other wnt disrupting mutations in Axin1

It is a very nice demonstration of how applied bioinformatics with functional follow up can maximally utilise publically available data to generate clinically translatable and relevant results. I like the paper very much

→ We thank the Reviewer for his/her general comments and appreciation of the work.

Specific comments and recommendations

Major

I only have 1 major comment - the identified "escape" Axin1 mutations are not completely characterised or considered

1. The c2503delG mutation looks to me as though it may actually be a c.2499delT mutation. This could be clarified with single strand cloning (using PGEMT or something similar). A c2499delT mutation is not listed in COSMIC, although this doesn't necessarily matter.
 → Indeed, looking at the Sanger sequencing electropherogram, T2499 seems to be heterozygous with G. However, this double signal could also reflect background noise in the electropherogram. We therefore checked carefully the exome sequencing data and found that all 291 reads covering nucleotide 2499 in VACO6_R cells displayed a T, ruling out the hypothesis that the single nucleotide deletion could have occurred in this position. Conversely, the c.2503delG was supported by 90 out of 357 reads, corresponding to an estimated allelic frequency of 25%. Moreover, Sanger sequencing electropherograms from VACO6_R clones (see reply at point 4 and Appendix Fig S6) displayed much lower background noise at position 2499, which resulted wild-type in all clones.
2. The identified mutations appear to be in polyC and poly G repeat tracts, was there any evidence of acquired microsatellite instability to explain this? Polymerase slippage could also be excluded with single strand cloning
 → Indeed, VACO6 cells are microsatellite instable. We reported this observation in the Results and Discussion – Section 2.
3. Any copy number changes in the VACO6R cells?
 → We had originally checked for possibly significant copy number changes, with negative results, and did not report it. Indeed, there seems to be a trisomy of the whole chromosome 8 and a heterozygous deletion from 13q21.39 to 13q31.1. This information is now reported in Results and Discussion, as follows: “Copy number analysis based on exome data only highlighted minor changes of no clear functional meaning in VACO6R cells: a trisomy of the whole chromosome 8 and a heterozygous deletion from 13q21.39 to 13q31.1.”
4. The authors have not completed single cell/clone analysis to determine whether these 2 widely spaced frameshift mutations are occurring in the same escaper cell clone. If this is the case why would a cell need to acquire 2 frameshift mutations when the first will suffice to truncate the protein? This should be considered.
 If the alternative scenario is true, and two different populations of the same escaper line have independently acquired a different Axin1 mutation this would be strong evidence for positive selection and an important functional role of this protein in Porcupine inhibition escape. Why this gene and not other frequently mutated Wnt disrupting mutations such as APC, Ctnnb1 should be considered and commented on.
 → This is a very interesting point. Indeed, a single frameshift mutation would abrogate Axin1 only partially: the remaining allele would still lead to expression of a functional Axin1 protein. In this view, the second frameshift mutation could result in complete loss of function. To further explore the problem, we isolated 14 independent clones from VACO6_R cells by limiting dilution. All clones were grown in the presence of 1 μM LGK-974, and subsequently underwent Sanger sequencing of the AXIN1 gene at the two mutation sites, plus western blot analysis of AXIN1 protein expression. We found that both frameshift mutations were present in all clones, each of them being heterozygous, which suggests that the vast majority VACO6_R cells emerged from one resistant subclone in which the two AXIN1 alleles were independently inactivated. Accordingly, all clones displayed complete loss of AXIN1 protein, as previously observed for VACO6R (Appendix Fig S6). Regarding the second comment, i.e. “Why this gene and not other frequently mutated Wnt disrupting mutations such as APC, Ctnnb1”, our tentative response is: “because that was the mutant gene already present in a subclone of VACO6 before selection”. However, we now have further data generated in response to Referee#3, showing that also APC downregulation or CTNNB1 activating mutations may render VACO6 cells resistant to porcupine inhibition.

Minor comments

How the CAF score was generated should be in the methods. It is too big a part of the story to require referral to a separate paper

→ We now briefly explain in Methods the procedures to calculate the CAF score, as follows: “To accurately trace the stromal content from gene expression profiles of bulk CRC tumor samples, we exploited three stromal signatures (for CAFs, leukocytes and endothelial cells) that we previously developed from sorted CRC cell subpopulations (Isella et al, 2015). Averaging the expression of genes in each signature yielded three stromal scores (CAF-score, Leuko-score and Endo-score, respectively), reporting the abundance of the three stromal cell populations in the sample.”

Page 2, fig S1. Is the correlation of RSPO expression with the leucocyte and endothelial score because these scores are simply surrogate markers of the volume of tumour stroma. Or do the endothelium and leucocytes act as a source of R-Spo?

→ If the endothelium or leucocytes act as a source of RSPO3, then those cases with high Endo- or Leuco-score, but low CAF-score, could still express higher-than average levels of RSPO3. Based on this reasoning, after excluding cases with RSPO3 fusion, we selected TCGA samples with high Endo- or Leuco- score (>70th percentile) but low CAF-score (<30th percentile) and compared RSPO3 mRNA levels with the remaining cases. For both scores, RSPO3 mRNA expression was indeed lower than average, as shown in the dot plots below. This confirms that Leuco- and Endo-scores represent weaker surrogate markers of stromal abundance, while endothelium and leucocytes do not seem to be a major source of RSPO3.

Page 2 Any idea what causes the high epithelial R-Spo score in the other 8/14 tumours without detected R-Spo fusions?

→ In the other 8/14 tumors, the source of RSPO3 mRNA is mostly stromal rather than epithelial, as these cases have high CAF-score, as now can be clearly seen in the new version of Figure 1A. We mentioned in the introduction that stromal cells have been identified as a source for RSPO3 expression in the intestine (Kabiri et al, 2014).

Page 3 'in' needed - and a novel PRPTK(ex13)-RPO3 (ex 2) fusion transcript IN SNU1411

→ We corrected the text.

Page 3 - last paragraph

A sentence setting out the differences between the ligand dependent Wnt activation caused by R-SPO fusions as apposed to the constitutive activation caused by APC for example, would be useful for the non-expert reader

→ The following sentence has been added: “The vast majority of CRC are affected by loss-of-function mutations in components of the destruction complex (e.g. APC) leading to accumulation of β -catenin and constitutive activation of Wnt target genes. RSPO3 instead promotes WNT pathway

activation by binding the LGR4/5 protein and neutralizing RNF43-mediated degradation of LRP5/6 receptor, enhancing therefore the activity of WNT ligands (de Lau et al, 2014)".

Page 4 para 1 - ...previously described for RSPO3 blockade. Not referenced
 → We now included the references in the text.

Page 4 para 1 - Last sentence isnt very clear. Do they mean that the long upstream coding sequence from PTPRK does not lessen the pathological activation of R-SPO3 expression?
 → We thank the reviewer for this comment. We corrected the sentence clarifying the concept: "This is particularly interesting for SNU1411 cells with the non canonical RSPO3 fusion, in which the long upstream coding sequence from PTPRK does not seem to lessen the pathological activation of WNT pathway driven by aberrant RSPO3 expression."

Figure 2C and D Label the cell line on the figure
 → The names of the cell lines are now displayed in Figures 2C and D.

Referee #3 (Remarks):

Comments on manuscript number: EMM-2016-06773

Title: Loss of Axin1 drives acquired resistance to WNT pathway blockade in colorectal cancers cells

A small, but clinically nonetheless relevant subgroup of colorectal cancers carries RSPO2/3 fusion genes that lead to aberrant activation of Wnt/ β -catenin signaling in a manner dependent on the availability of Wnt ligands. Cancers driven by RSPO2/3 fusion gene expression therefore provide a unique opportunity for therapeutic intervention through the inhibition of Wnt ligand production or function. The design and realization of such therapeutic approaches could be fostered enormously by the development of suitable in vitro models. As a first step towards this goal Gabriele Picco and co-workers develop an elegant bioinformatic approach that should allow for high confidence identification of colorectal tumor samples which show high level tumor-specific expression of RSPO3 and which harbor RSPO3 fusion genes. This approach is based on the processing of transcriptome data and involves filtering out interfering gene expression signals from contaminating stromal cells in order to identify outlier samples with RSPO3 overexpression. The major achievement of this study is the successful application of the "outlier expression" approach to identify two colorectal cancer cell lines with PTPRK-RSPO3 fusions. These cell lines are further established as valuable preclinical model systems by demonstrating in vitro and in vivo their sensitivity against an inhibitor of Porcupine, an acyltransferase which is essential for Wnt ligand production. The potential utility of the newly identified cell lines is additionally demonstrated through the identification of mutations in the AXIN1 gene that might be the underlying cause for acquired resistance against the Porcupine inhibitor used. This manuscript is well written, is easy to follow, and provides sufficient background information to convey the significance and the importance of the study also to a non-expert readership. The technical quality is high and the vast majority of the authors' conclusions are strongly supported by the analyses and data presented. In my opinion, there are only two aspects that require further elaboration: These are, first, the reliability of the identification process of cancers with RSPO2/3 fusion genes, and second, the potential mechanisms whereby cancers with RSPO2/3 fusions might acquire resistance against Wnt pathway inhibitors, specifically the role of AXIN1 mutations is in this context.

Specific comments

1. By looking at Figure 1A and Appendix Table1A it appears to me that one might have identified tumor samples with RPSO2/3 fusions without further bioinformatic analysis simply by picking the samples with high RSPO3 expression and a low CAF score. I suggest to individually label the 14 tumor samples within the red box in Figure 1A such that they can be tracked throughout the manuscript for example in Figure 1B, Appendix Figure 1A and Appendix Figure 1B. This would help to answer the question whether the same tumor samples are identified after stratification using the endothel and leukocyte gene expression signatures. Furthermore, are there false negatives, i. e. tumor samples with RSPO2/3-fusions that were not identified through the data processing pipeline developed by the authors?

→ Indeed, the samples highlighted as white and red circles in figure 1B are all those contained in the red box of Figure 1A, but this was not clear. We now modified Figure 1A and highlighted

individually the samples included in the red box, to enable their tracking in Figure 2B and also in Appendix Fig S1A and Appendix Fig S1B. In principle, we agree that a simple look at the plot in Figure 1A could allow “hand-picking” of the promising samples. However, the calculation of the RSPO3-CAF score allows non-arbitrary statistical selection of samples based on outlier expression. To define if TCGA samples not identified through our data processing pipeline could carry RSPO3 fusions, we further expanded the search for RSPO3 fusions to 12 additional TCGA RNAseq samples expressing decreasing levels of RSPO3. In particular, for any expression level, we selected the samples with the lowest CAF score (Appendix Fig S3), so that at least a fraction of the RSPO3 reads could theoretically derive from epithelial cells. Samples with further lower levels were not explored because the limited number of RNAseq reads covering RSPO3 was estimated not to allow detection of a fusion transcript. None of the additionally analyzed samples was found to carry a RSPO3 fusion transcript, suggesting that our approach saturated the dataset and captured all the samples carrying RSPO3 rearrangements. These results are now included in the manuscript.

2. The difference between the RSPO3 and the (RSPO3-CAF) Z-score for the GSE14333 data set is much less pronounced and clear-cut (Appendix Figure 2) when compared to the analysis of the TCGA data set shown in Figure 1B. So what is the significance of this analysis for the identification with high confidence of tumor samples with RSPO2/3 fusions? Which of the GSE14333 samples are actually positive for RSPO2/3 fusion genes? If this information is not available for GSE14333, how would the analysis for the tumors samples described by Seshagiri and co-authors (Nature 488, 660-664, 2012) look like?

→ Unfortunately RSPO2/3 fusion data are not available for GSE14333. Seshagiri and colleagues published RSPO2/3 expression and fusion data, but not global expression profiles, required to calculate the CAF-score. Access to raw RNAseq data is restricted, therefore we could not proceed with this analysis. In any case, also the Seshagiri data are based on RNAseq, not microarray. To further validate results already presented in Appendix Figure 2, we have assembled a larger 2140-sample CRC microarray dataset. We have calculated the three stromal scores (CAF, Endo and Leuco) to subsequently compare them with RSPO3 mRNA expression. This analysis confirms our results obtained in CRC TCGA RNAseq dataset, strongly supporting the evidence that the subtraction of CAF-score from RSPO3 mRNA expression clearly highlights the presence of a subgroup of CRC with outlier’s level of epithelial RSPO3 mRNA expression. These samples are high-confidence candidates to carry RSPO3 rearrangements. We now reported these results in the manuscript and we updated Appendix Fig S2 accordingly.

3. The authors ponder whether the long upstream portion from the PRPTK gene in the PTPRK(ex13)-RSPO3(ex2) fusion interferes with RSPO3 function. But does this fusion protein actually exist or is it somehow processed to release the RSPO3 portion? This could be addressed by Western blotting to analyze the expression of the RSPO3 fusion protein and its apparent molecular weight.

→ To address this task, we have tried various commercially available antibodies but unfortunately none of these was functional and reliable, with many background bands in cell extracts and in supernatants. We also tried to silence the RSPO3 transcript by shRNA in VACO6 and SNU1411 cells and despite the fusion transcript was significantly downregulated as seen by RT-PCR, no WB band potentially corresponding to the chimeric RSPO3 protein or to a fragment was downregulated. Indeed, detection of RSPO3 by western blot seems a common problem. Accordingly, to the best of our knowledge, the only published western blot of RSPO3 was performed after exogenous expression of tagged RSPO3 fusion transcripts (Seshagiri et al, 2012).

Despite this, the overexpression of the aberrant RSPO3 transcript and the striking addiction of SNU1411 to porcupine blockade strongly prompt us to speculate that the long upstream portion from the PRPTK gene in the PTPRK(ex13)-RSPO3(ex2) fusion does not interfere with RSPO3 function.

4. From the manuscript it does not become clear whether the AXIN1 mutations are the only newly acquired genetic alterations in VACO6R cells. This needs to be clarified because if not, the significance of AXIN1 inactivation as resistance mechanism would be dubitable despite the phenocopy by the AXIN1 knockdown. Along the same line, can re-expression of AXIN1 restore sensitivity to LGK-974? Furthermore, it should be clearly shown that the RSPO3-fusion is still present and expressed in LGK-974-resistant cells.

→ To investigate if AXIN1 mutations were the only alteration acquired in VACO6_R cells, we carefully re-analyzed the exome sequencing data and failed to identify other mutated genes known

to be involved in WNT signaling. Indeed, a series of additional indels/mutations with high allelic frequency were detected (we now report them in Appendix Table 2). In the context of a MSI cell line, a large set of mutations at low allelic frequency is expected as a consequence of genetic drift. However, in this case, the number of concurrent mutations at high allelic frequency points to the occurrence of a separated clone, pre-existing to the selection process. We included this information and comment in the manuscript.

To confirm the causative role of AXIN1 loss in resistance to LGK974, we re-expressed the wild-type AXIN1 coding sequence in VACO6_R cells, to levels comparable to the parental VACO6 cells. Indeed, VACO6_R-AXIN1 cells displayed a reversion of the phenotype: reduction of basal WNT pathway activity and re-sensitization to LGK-974 (Figure EV1B-D).

Finally, we also verified by qRT-PCR that the RSPO3 fusion transcript is equally expressed in VACO6_R cells as in the parental counterparts (Figure EV1A).

We now mention all these results in the last section of the Results and Discussion.

5. After dose escalation for three months the authors isolate a LGK-974-resistant cell population and report the biallelic inactivation of AXIN1 in this population. Is there something special and perhaps unique about AXIN1 with respect to resistance against Porcupine inhibition? This could become important for the development of drugs that break secondary resistance. The authors need to provide information whether their resistant cell population is of monoclonal or polyclonal origin. Did AXIN1 inactivation occur repeatedly and independently in multiple cases? Is AXIN1 inactivation the only way to become insensitive towards LGK-974? The authors should knockdown APC and overexpress mutant β -catenin in VACO6 cells and compare this to AXIN1 knockdown with respect to LGK-974 sensitivity.

→ We thank the reviewer for these insightful comments. To address the clonality issue, we isolated 14 independent clones from VACO6_R cells by limiting dilution. All clones were grown in the presence of 1 μ M LGK-974, and subsequently underwent Sanger sequencing of the AXIN1 gene at the two mutation sites and western blot analysis of AXIN1 protein expression. We found that both frameshift mutations were present in all clones, each of them being heterozygous, which suggests that the vast majority VACO6_R cells emerged from one resistant subclone in which the two AXIN1 alleles were independently inactivated. Accordingly, all clones displayed complete loss of AXIN1 protein, as previously observed for VACO6_R (Appendix Fig S6).

To further investigate if AXIN1 inactivation is the only way for VACO6 cells to become resistant to LGK974, we transduced them with an activated β -catenin, which induced basal WNT pathway activation (Figure EV2B and EV3A) and impaired the response to PORCN inhibition by LGK974 and WNT-59 (Figure EV3B). Also transient APC downregulation by RNA interference induced activation of WNT signaling (Figure EV2B and EV3A) and resistance to LGK974 (Figure EV3B). Overall, these results confirm that also the more frequent alterations of WNT pathway components, APC loss and β -catenin activating mutation, can confer resistance to WNT pathway inhibition in CRC cells with RSPO3 rearrangements.

We now mention these results in the manuscript in the last section of the Results and Discussion.

6. From a therapeutic point of view it would be interesting and highly relevant to know whether AXIN1 mutation/knockdown confers resistance specifically against LGK-974. How about resistance against other Porcupine and Wnt pathway inhibitors? Did VACO6_R cells become resistant against other drugs, unrelated to the Wnt pathway, as well?

→ To address the first point, we assessed sensitivity of VACO6 and VACO6_R cells to additional WNT pathway inhibitors: the alternative porcupine inhibitor WNT-C59 and the tankyrase inhibitor XAV939. While VACO6 parental cells were found to be markedly sensitive to WNT-C59 and XAV939, VACO6_R were completely resistant to both compounds (Appendix Fig S7). Altogether, these data confirm that CRC cells carrying RSPO3 fusions are addicted to WNT pathway blockade and that AXIN1 loss reverts this dependence. To assess if VACO6_R acquired resistance to additional therapeutic compounds, unrelated to the WNT pathway, we tested their sensitivity to two chemotherapeutic agents commonly used to treat CRC patient (5FU and SN38) and Pevonedistat, a NEDD-8 inhibitor recently identified as promising therapeutic strategy for CRCs, to which VACO6 are markedly sensitive (Picco et al., JNCI 2017) (Appendix Fig S7). The observed response of VACO6_R cells to these three compounds was superimposable to that of their parental counterparts. We now include these results, in the last section of the Results and Discussion.

7. The authors show that AXIN2 is expressed in VACO6 cells and that AXIN2 is further upregulated in VACO6_R cells. AXIN1 and AXIN2 are considered to be functionally redundant and

fully interchangeable. In view of this, how do the authors explain that elevated AXIN2 does not compensate the loss of AXIN1? What are the expression ratios of AXIN1:AXIN2? Is AXIN2 expressed at the protein level?

→ It is well known that AXIN2 is involved in a negative feedback loop; indeed, we agree with the reviewer that its upregulation upon AXIN1 loss could mitigate WNT pathway activation. However, not always AXIN1 and AXIN2 are functionally redundant, see e.g.

<http://www.ncbi.nlm.nih.gov/pubmed/25866367>, where “*retroviral-mediated overexpression of Axin2 was unable to compensate for knockdown of Axin1*”.

It should also be noted that AXIN2 upregulation in VACO6R cells, though statistically significant, is only about 1.5-fold, while AXIN1 is completely lost.

We now discuss this point in the last section of the Results and Discussion.

Regarding the question of the AXIN1:AXIN2 ratio, absolute mRNA expression levels measured by microarray are not completely reliable, due to differences in probe hybridization efficiency and specificity. The situation is even worse for protein quantification, affected by sensitivity and specificity of the antibody, plus the enzymatic amplification of the signal. With this caveat in mind, looking at the mRNA expression of AXIN1 and AXIN2 across the 151 CRC cell line dataset, we observed that the AXIN2 signal is by average 4-fold higher than AXIN1; interestingly, in VACO6, the ratio is less than 2-fold. This indicates that, in VACO6, the contribution of AXIN1 to the total AXIN levels is higher than average. We however do not consider this information reliable enough to include it in the manuscript.

8. The title should be more precise and avoid any kind of generalization that could be misleading. It should be clearly mentioned that the study applies to colorectal cancers with RSPO2/3 fusion genes and the interference with pathway activity at the level of Wnt ligand production/action.

→ We modified the title in relation to the first point. As we now have additional data also relative to a more downstream tankyrase inhibitor, specifying the WNT ligand production/action is not necessary.

9. The Chartier et al 2015 and Madan et al 2015 references are incomplete in the bibliography.

→ We updated the bibliography entering the missing details.

2nd Editorial Decision

06 December 2016

Thank you for the submission of your revised manuscript to EMBO Molecular Medicine.

We have now received the enclosed reports from the reviewers that were asked to re-assess it. As you will see, while reviewers 1 and 2 are now globally supportive, reviewer 3, while acknowledging that the manuscript is improved, is still not satisfied. His/her main concern is that the clinical relevance of your finding (quite important given the type of manuscript) is still limited due to the as yet not clarified issue as to whether the RSPO2/3 identification process is thorough enough to detect most if not all RSPO2/3 fusions or just touches the tip of a virtual iceberg. The reviewer suggests that a more appropriate analysis should be performed and is within your reach. S/he also lists a couple of other points. Although I will not be asking you to provide additional experimentation at this stage (although further analysis would be welcome), I do suggest you thoroughly address these concerns with a rebuttal and by appropriately amending the manuscript as necessary. Provided your response is exhaustive, I might be able to make an editorial decision on your manuscript.

Please also carry out the following amendments to accelerate the process should your manuscript be considered for acceptance:

- 1) The appendix file should feature a table of contents on the first page.
- 2) Appendix table callouts should be as follows: Appendix Table S1, Appendix Table S2, etc.
- 3) As per our Author Guidelines, the description of all reported data that includes statistical testing must state the name of the statistical test used to generate error bars and P values, the number (n) of independent experiments underlying each data point (not replicate measures of one sample), and the actual P value for each test (not merely 'significant' or 'P < 0.05').

4) We encourage the publication of source data, particularly for electrophoretic gels and blots, with the aim of making primary data more accessible and transparent to the reader. Would you be willing to provide a PDF file per figure that contains the original, uncropped and unprocessed scans of all or at least the key gels used in the manuscript? The PDF files should be labeled with the appropriate figure/panel number, and should have molecular weight markers; further annotation may be useful but is not essential. The PDF files will be published online with the article as supplementary "Source Data" files. If you have any questions regarding this just contact me.

5) Every published paper includes a 'Synopsis' to further enhance discoverability. Synopses are displayed on the journal webpage and are freely accessible to all readers. They include a short standfirst as well as 2-5 one sentence bullet points that summarise the paper. Please provide the synopsis including the short list of bullet points that summarise the key NEW findings. The bullet points should be designed to be complementary to the abstract - i.e. not repeat the same text. We encourage inclusion of key acronyms and quantitative information. Please use the passive voice. Please attach this information in a separate file or send them by email, we will incorporate it accordingly. You are also welcome to suggest a striking image or visual abstract to illustrate your article. If you do please provide a jpeg file 550 px-wide x 400-px high.

Please submit your revised manuscript within two weeks. I look forward to seeing a revised form of your manuscript as soon as possible.

I look forward to reading a new revised version of your manuscript as soon as possible.

***** Reviewer's comments *****

Referee #1 (Remarks):

Maybe my initial statement that mutations in downstream regulators "might have been expected" was too harsh and somehow underrates the value of this well performed study.

Referee #2 (Remarks):

The authors have responded to all my comments to my satisfaction. The hypothesis that a preexisting, biallelic Axin1 mutant clone emerges through selective pressure in the VACOR cells is biologically plausible, especially in an MSI setting. I continue to like the paper very much!

There is a misspelling of frequent in the last paragraph of the discussion

Referee #3 (Remarks):

Comments on manuscript number: EMM-2016-06773-V2

Title: Loss of Axin1 drives acquired resistance to WNT pathway blockade in colorectal cancers cells carrying RSPO3 fusions.

As in its original version, the study by G. Picco and colleagues follows two main goals. On the one hand, the authors aim to reliably and sensitively identify colon tumors with RSPO2/3 fusions based on transcriptome data. On the other hand, they aim to generate a cell-line based model to evaluate treatment options for tumors with these fusion transcripts and to anticipate potential resistance mechanisms that might develop in response to therapy. Concerning the second aim, the authors performed a substantial amount of additional experiments and added quite a bit of new and valuable information to the revised version of the manuscript. Overall, the manuscript underwent major changes and improved significantly. The main findings and conclusions were substantiated and the work has been put into a wider perspective with respect to its medical and therapeutic relevance. Specifically, the authors clarified that the observed resistance of VACO6R cells against LGK-974 is the result of a singular, clonal event. This notwithstanding, they add as new information that also APC loss-of-function and β -catenin gain-of-function could lead to resistance against LGK-974. In

addition, the authors demonstrated Wnt pathway specificity of the resistance mechanism in VACO6R cells. Moreover, despite a large number of additional mutations present in the VACO6R background, AXIN1 mutations are firmly established as causative for acquired resistance. For this, the authors performed the AXIN1 rescue which restored sensitivity against LGK-974. Accordingly, the authors nicely removed most of my previous concerns. The one issue that was not completely solved concerns the reliability and sensitivity of their data processing pipeline for the identification of tumors and cell lines with RSPO2/3 fusion transcripts. The GSE14333 data set was replaced by a much larger collection of transcriptome data which indeed allowed pinpointing many more candidates for carriers of RSPO2/3 fusion transcripts. However, the crucial information, namely which of these actually express such fusions is still missing. Aside from this, in order to find tumors with RSPO2/3 fusions transcripts that escaped detection by their outlier approach, the authors performed a rather restricted analysis and selected a small sample of tumors for additional inspection. The rationale for this restriction is not clear to me. In my opinion, a completely unbiased search for samples with RSPO2/3 fusions transcripts across an entire data set would have been feasible and more informative. After all, the authors have established and applied the tools for this kind of search and they have available the transcriptome data from the TCGA collection and their 151 cell line collection both of which would have served the purpose.

Minor issues:

- The authors should update the Methods section and include the relevant information for the other Wnt pathway inhibitors and additional drugs that were used during the revision.
- Response to previous comment 7, AXIN1/AXIN2 ratios in VACO6 cells: the authors' concerns regarding microarray- and antibody-based comparative quantifications are certainly valid. However, a qRT-PCR measurement of the relative expression levels of AXIN1 and AXIN2 transcripts in VACO6 and VACO6R cells could be informative.

2nd Revision - authors' response

18 December 2016

Referee #1 (Remarks):

Maybe my initial statement that mutations in downstream regulators "might have been expected" was too harsh and somehow underrates the value of this well performed study.

Referee #2 (Remarks):

The authors have responded to all my comments to my satisfaction. The hypothesis that a preexisting, biallelic Axin1 mutant clone emerges through selective pressure in the VACO6R cells is biologically plausible, especially in an MSI setting.

I continue to like the paper very much!

There is a misspelling of frequent in the last paragraph of the discussion

→ We corrected the text accordingly.

Referee #3 (Remarks):

Comments on manuscript number: EMM-2016-06773-V2

Title: Loss of Axin1 drives acquired resistance to WNT pathway blockade in colorectal cancers cells carrying RSPO3 fusions.

As in its original version, the study by G. Picco and colleagues follows two main goals. On the one hand, the authors aim to reliably and sensitively identify colon tumors with RSPO2/3 fusions based on transcriptome data. On the other hand, they aim to generate a cell-line based model to evaluate treatment options for tumors with these fusion transcripts and to anticipate potential resistance mechanisms that might develop in response to therapy. Concerning the second aim, the authors performed a substantial amount of additional experiments and added quite a bit of new and valuable information to the revised version of the manuscript. Overall, the manuscript underwent major changes and improved significantly. The main findings and conclusions were substantiated and the work has been put into a wider perspective with respect to its medical and therapeutic relevance. Specifically, the authors clarified that the observed resistance of VACO6R cells against LGK-974 is the result of a singular, clonal event. This notwithstanding, they add as new information that also

APC loss-of-function and β -catenin gain-of-function could lead to resistance against LGK-974. In addition, the authors demonstrated Wnt pathway specificity of the resistance mechanism in VACO6R cells. Moreover, despite a large number of additional mutations present in the VACO6R background, AXIN1 mutations are firmly established as causative for acquired resistance. For this, the authors performed the AXIN1 rescue which restored sensitivity against LGK-974. Accordingly, the authors nicely removed most of my previous concerns. The one issue that was not completely solved concerns the reliability and sensitivity of their data processing pipeline for the identification of tumors and cell lines with RSPO2/3 fusion transcripts. The GSE14333 data set was replaced by a much larger collection of transcriptome data which indeed allowed pinpointing many more candidates for carriers of RSPO2/3 fusion transcripts. However, the crucial information, namely which of these actually express such fusions is still missing. Aside from this, in order to find tumors with RSPO2/3 fusions transcripts that escaped detection by their outlier approach, the authors performed a rather restricted analysis and selected a small sample of tumors for additional inspection. The rationale for this restriction is not clear to me. In my opinion, a completely unbiased search for samples with RSPO2/3 fusions transcripts across an entire data set would have been feasible and more informative. After all, the authors have established and applied the tools for this kind of search and they have available the transcriptome data from the TCGA collection and their 151 cell line collection both of which would have served the purpose.

→ Regarding the possibility to search systematically for RSPO fusions in our 151 cell lines, it should be noted that these cells have been expression-profiled with microarrays, not RNAseq. Of course a dedicated RNAseq profiling was then performed for SNU1411 and VACO6 cells, but systematic fusion search in the remaining lines was not possible.

In the case of TCGA data, our estimated time for downloading ten TCGA RNAseq samples and processing them by the multiple fusion detection algorithms envisioned by our pipeline was around 10 days, and varying depending on bandwidth and computational resources available. Therefore, a complete analysis would have required much more than three months to be performed properly. For this reason we concentrated on the subset displaying RSPO3 expression not completely justified by stromal gene expression, as described in the revised manuscript.

To obtain further available information on TCGA RNAseq data, we found a useful resource: the TCGA Fusion Gene Data Portal (<http://54.84.12.177/PanCanFusV2/>), that provides the results of a systematic fusion transcript analysis across many tumour types from TCGA, including CRC (<https://www.ncbi.nlm.nih.gov/pubmed/25500544>). Searching this resource for RSPO3 fusions in colorectal cancer (312 colon + 95 rectum adenocarcinoma samples analysed) yielded 7 cases with RSPO3 fusion, all canonical (PTPRK-RSPO3). Of these, three were not included in the 450-sample TCGA RNAseq dataset that we used for outlier expression analysis. The remaining four were all among those that our pipeline identified as high confidence candidates and found to contain the PTPRK-RSPO3 fusion, based on multiple fusion detection algorithms. Interestingly, we found two additional samples with PTPRK-RSPO3 fusion, not included in the TCGA Fusion Gene Data Portal. It is not possible for us to know whether those samples were included in their analysis or not, because the database only shows samples with a detected fusion. This result, based on an independent analysis, demonstrates the efficacy of our strategy to prioritize samples for RSPO3 fusion search, greatly reducing the required effort while maintaining optimal sensitivity. We have now included this information in the manuscript.

Minor issues:

- The authors should update the Methods section and include the relevant information for the other Wnt pathway inhibitors and additional drugs that were used during the revision.

→ We updated the Methods section as suggested.

- Response to previous comment 7, AXIN1/AXIN2 ratios in VACO6 cells: the authors' concerns regarding microarray- and antibody-based comparative quantifications are certainly valid. However, a qRT-PCR measurement of the relative expression levels of AXIN1 and AXIN2 transcripts in VACO6 and VACO6R cells could be informative.

→ In our opinion, this is a “catch-22” situation, where a well-known negative feedback loop would remain confirmed irrespectively of the results of a detailed analysis of AXIN1 vs AXIN2 levels. The system would anyway reach an equilibrium, whereby, in the absence of AXIN1, WNT signaling would raise, and so consequently would AXIN2, that in turn would favour beta-catenin destabilization, leading to WNT signaling downregulation, until the system reaches a new equilibrium in which both WNT signaling and AXIN2 are increased. The possible relevance of the proposed qPCR analysis is further turned down by the experiments in which we used the WNT pathway reporter GFP construct (Figure EV1 C). We transduced with this reporter VACO6,

VACO6_R and VACO6_R-Axin1FL cells, and found that GFP fluorescence is markedly increased in VACO6_R (lacking AXIN1), and returns to the initial activity upon transduction of VACO6_R with AXIN1 full-length. This finding is a much more solid demonstration of basal WNT pathway upregulation in VACO6_R cells than the AXIN2 qPCR data presented in the first version of the manuscript.

YOU MUST COMPLETE ALL CELLS WITH A PINK BACKGROUND
PLEASE NOTE THAT THIS CHECKLIST WILL BE PUBLISHED ALONGSIDE YOUR PAPER

Corresponding Author Name: Enzo Medico
Journal Submitted to: EMBO Molecular Medicine
Manuscript Number: EMM-2016-06773

Reporting Checklist For Life Sciences Editors (Rev. July 2015)

This checklist is used to ensure good reporting standards and to improve the reproducibility of published results. These guidelines are consistent with the Principles and Guidelines for Reporting Preclinical Research issued by the NIH in 2014. Please follow the journal's authorship guidelines in preparing your manuscript.

A- Figures

1. Data

The data shown in figures should satisfy the following conditions:

- the data were obtained and processed according to the field's best practice and are presented to reflect the results of the experiments in an accurate and unbiased manner.
- figure panels include only data points, measurements or observations that can be compared to each other in a scientifically meaningful way.
- graphs include clearly labeled error bars for independent experiments and sample sizes. Unless justified, error bars should not be shown for technical replicates.
- if $n \leq 5$, the individual data points from each experiment should be plotted and any statistical test employed should be justified
- Source Data should be included to report the data underlying graphs. Please follow the guidelines set out in the author ship guidelines on Data Presentation.

2. Captions

Each figure caption should contain the following information, for each panel where they are relevant:

- a specification of the experimental system investigated (eg cell line, species name).
- the assay(s) and method(s) used to carry out the reported observations and measurements
- an explicit mention of the biological and chemical entity(ies) that are being measured.
- an explicit mention of the biological and chemical entity(ies) that are altered/varied/perturbed in a controlled manner.
- the exact sample size (n) for each experimental group/condition, given as a number, not a range;
- a description of the sample collection allowing the reader to understand whether the samples represent technical or biological replicates (including how many animals, litters, cultures, etc.).
- a statement of how many times the experiment shown was independently replicated in the laboratory.
- definitions of statistical methods and measures:
 - common tests, such as t-test (please specify whether paired vs. unpaired), simple χ^2 tests, Wilcoxon and Mann-Whitney tests, can be unambiguously identified by name only, but more complex techniques should be described in the methods section;
 - are tests one-sided or two-sided?
 - are there adjustments for multiple comparisons?
 - exact statistical test results, e.g., P values $>$ but not P values $<$;
 - definition of 'center values' as median or average;
 - definition of error bars as s.d. or s.e.m.

Any descriptions too long for the figure legend should be included in the methods section and/or with the source data.

Please ensure that the answers to the following questions are reported in the manuscript itself. We encourage you to include a specific subsection in the methods section for statistics, reagents, animal models and human subjects.

In the pink boxes below, provide the page number(s) of the manuscript draft or figure legend(s) where the information can be located. Every question should be answered. If the question is not relevant to your research, please write NA (non applicable).

USEFUL LINKS FOR COMPLETING THIS FORM

http://www.antibodypedia.com	Antibodypedia
http://1degreebio.org	1DegreeBio
http://www.equator-network.org/reporting-guidelines/improving-bio-science-research-reporting-the-art	ARRIVE Guidelines
http://grants.nih.gov/grants/olaw/olaw.htm	NIH Guidelines in animal use
http://www.mrcc.ac.uk/OurResearch/Ethicsresearchguidance/Useofanimals/index.htm	MRC Guidelines on animal use
http://ClinicalTrials.gov	Clinical Trial registration
http://www.consort-statement.org	CONSORT Flow Diagram
http://www.consort-statement.org/checklists/view/32-consort/56-title	CONSORT Check List
http://www.equator-network.org/reporting-guidelines/reporting-recommendations-for-tumour-marker	REMARK Reporting Guidelines (marker prognostic studies)
http://datadryad.org	Dryad
http://figshare.com	Figshare
http://www.ncbi.nlm.nih.gov/gap	dbGAP
http://www.ebi.ac.uk/ega	EGA
http://biomodels.net/	Biomodels Database
http://biomodels.net/miriam/	MIRIAM Guidelines
http://jil.biochem.sun.ac.za	RNS Online
http://dx.doi.org/10.1002/biot/biosecurity_documents.html	Biosecurity Documents from NIH
http://www.selectagents.gov/	List of Select Agents

B- Statistics and general methods

Please fill out these boxes (Do not worry if you cannot see all your text once you press return)

1.a. How was the sample size chosen to ensure adequate power to detect a pre-specified effect size?	NA
1.b. For animal studies, include a statement about sample size estimate even if no statistical methods were used.	Page 10.
2. Describe inclusion/exclusion criteria if samples or animals were excluded from the analysis. Were the criteria pre-established?	No samples or animals were excluded from any analysis.
3. Were any steps taken to minimize the effects of subjective bias when allocating animals/samples to treatment (e.g. randomization procedure)? If yes, please describe.	Page 10.
For animal studies, include a statement about randomization even if no randomization was used.	Page 10.
4.a. Were any steps taken to minimize the effects of subjective bias during group allocation or/and when assessing results (e.g. blinding of the investigator)? If yes please describe.	No blinding was done. Page 10.
4.b. For animal studies, include a statement about blinding even if no blinding was done.	Page 10.
5. For every figure, are statistical tests justified as appropriate?	Pages 9-10
Do the data meet the assumptions of the tests (e.g., normal distribution)? Describe any methods used to assess it.	Pages 9-10
Is there an estimate of variation within each group of data?	YES, standard error of the mean (SEM) was calculated within each experimental group. Page 17, legend Figure 2.
Is the variance similar between the groups that are being statistically compared?	YES. Figure 2

C- Reagents

6. To show that antibodies were profiled for use in the system under study (assay and species), provide a citation, catalog number and/or clone number, supplementary information or reference to an antibody validation profile. e.g., Antibodypedia (see link list at top right), 1DegreeBio (see link list at top right).	Page 11.
7. Identify the source of cell lines and report if they were recently authenticated (e.g., by STR profiling) and tested for mycoplasma contamination.	Page 7.

*for all hyperlinks, please see the table at the top right of the document

D- Animal Models

8. Report species, strain, gender, age of animals and genetic modification status where applicable. Please detail housing and husbandry conditions and the source of animals.	Page 10.
9. For experiments involving live vertebrates, include a statement of compliance with ethical regulations and identify the committee(s) approving the experiments.	Page 10.
10. We recommend consulting the ARRIVE guidelines (see link list at top right) (PLoS Biol. 8(6), e1000412, 2010) to ensure that other relevant aspects of animal studies are adequately reported. See author guidelines, under 'Reporting Guidelines'. See also: NIH (see link list at top right) and MRC (see link list at top right) recommendations. Please confirm compliance.	YES

E- Human Subjects

11. Identify the committee(s) approving the study protocol.	NA
12. Include a statement confirming that informed consent was obtained from all subjects and that the experiments conformed to the principles set out in the WMA Declaration of Helsinki and the Department of Health and Human Services Belmont Report.	NA
13. For publication of patient photos, include a statement confirming that consent to publish was obtained.	NA
14. Report any restrictions on the availability (and/or on the use) of human data or samples.	NA
15. Report the clinical trial registration number (at ClinicalTrials.gov or equivalent), where applicable.	NA
16. For phase II and III randomized controlled trials, please refer to the CONSORT flow diagram (see link list at top right) and submit the CONSORT checklist (see link list at top right) with your submission. See author guidelines, under 'Reporting Guidelines'. Please confirm you have submitted this list.	NA
17. For tumor marker prognostic studies, we recommend that you follow the REMARK reporting guidelines (see link list at top right). See author guidelines, under 'Reporting Guidelines'. Please confirm you have followed these guidelines.	NA

F- Data Accessibility

18. Provide accession codes for deposited data. See author guidelines, under 'Data Deposition'. Data deposition in a public repository is mandatory for: a. Protein, DNA and RNA sequences b. Macromolecular structures c. Crystallographic data for small molecules d. Functional genomics data e. Proteomics and molecular interactions	NA
19. Deposition is strongly recommended for any datasets that are central and integral to the study; please consider the journal's data policy. If no structured public repository exists for a given data type, we encourage the provision of datasets in the manuscript as a Supplementary Document (see author guidelines under 'Expanded View' or in unstructured repositories such as Dryad (see link list at top right) or Figshare (see link list at top right)).	NA

20. Access to human clinical and genomic datasets should be provided with as few restrictions as possible while respecting ethical obligations to the patients and relevant medical and legal issues. If practically possible and compatible with the individual consent agreement used in the study, such data should be deposited in one of the major public access-controlled repositories such as dbGAP (see link list at top right) or EGA (see link list at top right).	NA
21. As far as possible, primary and referenced data should be formally cited in a Data Availability section. Please state whether you have included this section. Examples: Primary Data Wetmore KM, Deutschbauer AM, Price MN, Arkin AP (2012). Comparison of gene expression and mutant fitness in Shewanella oneidensis MN-2. Gene Expression Omnibus GSE39462 Referenced Data Huang L, Brown AF, Lei M (2012). Crystal structure of the TRBD domain of TERT and the CRA/5 of TR. Protein Data Bank 4Q26 AP-MS analysis of human histone deacetylase interactions in CEM7 cells (2013). PRIDE: PX0000208	Pag 11.
22. Computational models that are central and integral to a study should be shared without restrictions and provided in a machine-readable form. The relevant accession numbers or links should be provided. When possible, standardized format (SBML, CellML) should be used instead of scripts (e.g. MATLAB). Authors are strongly encouraged to follow the MIRIAM guidelines (see link list at top right) and deposit their model in a public database such as Biomedelis (see link list at top right) or JWS Online (see link list at top right). If computer source code is provided with the paper, it should be deposited in a public repository or included in supplementary information.	NA

G- Dual use research of concern

23. Could your study fall under dual use research restrictions? Please check biosecurity documents (see link list at top right) and list of select agents and toxins (APHIS/CDIC) (see link list at top right). According to our biosecurity guidelines, provide a statement only if it could.	NA
---	----